# MorphoFeatures for unsupervised exploration of cell types, tissues, and organs in volume electron microscopy

**Valentyna Zinchenko**[1†], **Johannes Hugger**[2†], **Virginie Uhlmann**[2], **Detlev Arendt**[3*], **Anna Kreshuk**[1*]

[1]Cell Biology and Biophysics Unit, European Molecular Biology Laboratory (EMBL), Heidelberg, Germany; [2]European Bioinformatics Institute, European Molecular Biology Laboratory (EMBL), Cambridge, United Kingdom; [3]Developmental Biology Unit, European Molecular Biology Laboratory (EMBL), Heidelberg, Germany

**Abstract** Electron microscopy (EM) provides a uniquely detailed view of cellular morphology, including organelles and fine subcellular ultrastructure. While the acquisition and (semi-)automatic segmentation of multicellular EM volumes are now becoming routine, large-scale analysis remains severely limited by the lack of generally applicable pipelines for automatic extraction of comprehensive morphological descriptors. Here, we present a novel unsupervised method for learning cellular morphology features directly from 3D EM data: a neural network delivers a representation of cells by shape and ultrastructure. Applied to the full volume of an entire three-segmented worm of the annelid *Platynereis dumerilii*, it yields a visually consistent grouping of cells supported by specific gene expression profiles. Integration of features across spatial neighbours can retrieve tissues and organs, revealing, for example, a detailed organisation of the animal foregut. We envision that the unbiased nature of the proposed morphological descriptors will enable rapid exploration of very different biological questions in large EM volumes, greatly increasing the impact of these invaluable, but costly resources.

**\*For correspondence:**
arendt@embl.de (DA);
anna.kreshuk@embl.de (AK)

[†]These authors contributed equally to this work

**Competing interest:** The authors declare that no competing interests exist.

## Editor's evaluation

This paper marks a fundamental advance in reconstruction of volume EM images, by introducing the automatic assignment of cell types and tissues. This task has previously been done manually, resulting in a serious bottleneck in reconstruction, but the authors present compelling evidence that in at least some cases, automatic and semi-automatic techniques can match or better human assignment of cell and tissue types. These results will be of interest to almost all groups doing EM reconstruction, as they can speed up cell type assignment when the cell types are known, and provide an initial cell type and tissue classification when they are not.

## Introduction

Development of multicellular organisms progressively gives rise to a variety of cell types, with differential gene expression resulting in different cellular structures that enable diverse cellular functions. Structures and functions together represent the phenotype of an organism. The characterisation of cell types is important to understand how organisms are built and function in general (*Arendt, 2008*). Recent advances in single-cell sequencing have allowed the monitoring of differential gene expression in the cell types that make up the tissues and organs of entire multicellular organisms and led to the recognition of distinct regulatory programmes driving the expression of unique sets of

cell type-specific effector genes (*Hobert and Kratsios, 2019*; *Sebé-Pedrós et al., 2018a*; *Sebé-Pedrós et al., 2018b*; *Musser et al., 2021*; *Fincher et al., 2018*; *Siebert et al., 2019*). These distinct genetic programmes make the identity and thus define cell types (*Arendt et al., 2016*; *Tanay and Sebé-Pedrós, 2021*). The genetic individuation of cell types then translates into phenotypic differences between cells, and one of the most intriguing questions is to determine this causation for each cell type. Only then will we understand their structural and functional peculiarities and intricacies and learn how evolutionary cell type diversification has sustained organismal morphology and physiology. As a prerequisite towards this aim, we need a comprehensive monitoring and comparison of cellular phenotypes across the entire multicellular body, which can then be linked to the genetically defined cell types.

Recent efforts have focused on large-scale high-resolution imaging to capture the morphological uniqueness of cell types. For the first time, the rapid progress in volume EM allows this for increasingly large samples, including organs and entire smaller animals, with nanometer resolution (*Bae et al., 2021*; *Vergara et al., 2021*; *Zheng et al., 2018*; *Scheffer et al., 2020*; *Cook et al., 2019*; *Verasztó et al., 2020*). Supported by automated cell segmentation pipelines (*Heinrich et al., 2021*; *Macrina et al., 2021*; *Pape et al., 2017*; *Müller et al., 2021*), such datasets allow characterisation and comparison of cellular morphologies, including membrane-bound and membraneless organelles and inclusions, at an unprecedented level of detail for thousands of cells simultaneously (*Turner et al., 2022*; *Vergara et al., 2021*). A notable challenge of such large-scale studies is the analysis of massive amounts of imaging data that can no longer be inspected manually. The latest tools for visual exploration of multi-terabyte 3D data allow for seamless browsing through the regions of interest (*Pietzsch et al., 2015*; *Maitin-Shepard et al., 2021*), given such regions are specified and contain a limited number of cells. However, if the goal is to characterise and/or compare morphologies of most or all cells in the dataset, one has to be able to automatically assign each cell an unbiased comprehensive morphological description - a feature vector with values representing the morphological properties of the cell in the most parsimonious manner.

The need for automation and quantitative evaluation has triggered the design of complex engineered features that capture specific cell characteristics (*Barad et al., 2022*). However, such features often have to be manually chosen and mostly provide only a limited, targeted view of the overall morphology. To alleviate this problem, intricate modelling pipelines for cellular morphology quantification have been developed (*Ruan et al., 2020*; *Driscoll et al., 2019*; *Phillip et al., 2021*); still, these are often data- and task-specific. For unbiased monitoring and comparison of cellular morphologies, broadly applicable automated pipelines are missing. They should, in addition, be capable of extracting comprehensive representations that are not biased by manual feature selection, the end task, or the type of data used. We expect that developing such techniques will not only speed up the process of scientific discovery in big volume imaging datasets but also facilitate detecting unusual and underrepresented morphologies.

In the last decade, artificial neural networks have been shown to exceed manually engineered pipelines in extracting comprehensive descriptors for various types of data and to be particularly suited for the extraction of visual features (*Krizhevsky et al., 2012*; *Falk et al., 2019*; *Simonyan and Zisserman, 2014*; *Girshick et al., 2014*). Moreover, the rise of so-called self-supervised training methods has removed the necessity for generating massive amounts of manual annotations previously needed to train a network. Besides the obvious advantage of being less labour-intensive, self-supervised methods do not optimise features for a specific end-task but instead produce descriptors that have been shown to be useful in a variety of downstream tasks, such as image classification, segmentation, and object detection (*He et al., 2020*; *Van den Oord et al., 2018*; *Tian et al., 2020*). Such methods have already shown to be successful for phenotyping cells in image-based profiling (*Lu et al., 2019*; *Lafarge et al., 2019*) and for describing local morphology of patches or subcompartments within neural cells in 3D electron microscopy (EM) of brain volumes (*Huang et al., 2020*; *Schubert et al., 2019*). Building on the latest achievements, we aim to expand the feature extraction methods to automated characterisation of cellular morphology at the whole animal level.

Here, we present the first framework for the fully unsupervised characterisation of cellular shapes and ultrastructures in a whole-body dataset for an entire animal. We apply this new tool to the fully segmented serial block-face EM (SBEM) volume of the 6 days post fertilisation young worm of the nereid *Platynereis dumerilii*, comprising 11,402 mostly differentiated cells with distinct morphological

properties (*Vergara et al., 2021*). We show that our method yields morphological descriptors - *MorphoFeatures* - that are in strong agreement with human perception of morphological similarity and quantitatively outperform manually defined features on cell classification and symmetric partner detection tasks. We further illustrate how our features can facilitate the detection of cell types by morphological means, via similarity-based clustering in the MorphoFeature vector space, sorting the cells into morphologically meaningful groups that show high correlation with genetically defined types such as muscle cells and neurons. Our pipeline also allows for the characterisation of rare cell types, such as enteric neurons and rhabdomeric photoreceptors, and detects distinct cell populations within the developing midgut. Finally, defining feature vectors that also represent the morphology of immediate neighbours, we group cells into tissues and also obtain larger groupings that represent entire organs. We show that such neighbourhood-based *MorphoContextFeatures* clustering repro- duces manually annotated ganglionic nuclei in the annelid brain and represents a powerful tool to automatically and comprehensively detect the distinct tissues that belong to and make up the foregut of the nereid worm - including highly specialised and intricate structures such as the neurosecretory infracerebral gland (*Baskin, 1974*; *Hofmann, 1976*; *Golding, 1970*) and the axochord that has been likened to the chordate notochord (*Lauri et al., 2014*). We further show that such morphologically defined tissues and organs correlate with cell type and tissue-specific gene expression. Our work thus sets the stage for linking genetic identity and structure-function of cell types, tissues, and organs across an entire animal.

The MorphoFeatures for the *Platynereis* dataset, as well as the code to generate and analyse them are available at https://github.com/kreshuklab/MorphoFeatures, (*Zinchenko, 2023* copy archived at swh:1:rev:f13d505f68e0dc08bd5fed9121ee56e45b4bd6ac).

## Results
### Unsupervised deep-learning extracts extensive morphological features
Our pipeline has been designed to extract morphological descriptors of cells (MorphoFeatures) from EM data. It requires prior segmentation of all the cells and nuclei of interest. The pipeline utilises segmentation masks to extract single cells/nuclei and represents their morphology as a combination of three essential components: shape, coarse, and fine texture (*Figure 1A*). We train neural networks to represent these components independently, using point clouds as input for shape, low-resolution raw data for coarse texture with sufficient context and small, high-resolution crops of raw data for fine texture. Separating these components ensures they are all represented in the final features and facilitates further interpretation. The features are combined to form one feature vector for each cell of the animal.

A common way to train a neural network for extracting relevant features is to use some prior infor- mation about samples in the dataset as a source of supervision. For example, to extract morpholog- ical representations of cells, one could use a complementary task of predicting cell lines (*Yao et al., 2019*; *Doan et al., 2020*; *Eulenberg et al., 2017*), experimental conditions the cells are coming from *Caicedo et al., 2018*, or classification of proteins fluorescently labelled in the cells (*Kobayashi et al., 2021*). However, such metadata is mostly unavailable for EM volumes, and manual annotation might become infeasible with increasing data size. Moreover, exploratory studies are often aimed at discovering unusual morphology, new cell types, or tissues. In this case, defining supervision is not only difficult but might also bias the exploration towards the 'proxy' groups used for supervision. To enable immediate data exploration even for datasets where direct supervision is hard to define and the ground truth cannot easily be obtained, we developed a fully unsupervised training pipeline that is based on two complementary objectives (*Figure 1B and C*). The first is an autoencoder reconstruction loss, where a network extracts a low-dimensional representation of each cell and then uses this repre- sentation to reconstruct back the original cell volume. This loss encourages the network to extract the most comprehensive description. The second objective is a contrastive loss that ensures the feature vectors extracted from two similarly looking cells (positive samples) are closer to each other than to feature vectors extracted from more dissimilar cells (negative samples). Since we do not know in advance which samples can be considered positive for our dataset, we are using slightly different views of the same cell, generated by applying realistic transformations to cell volumes (see Materials and methods). The combination of these two losses encourages the learned features to retain the

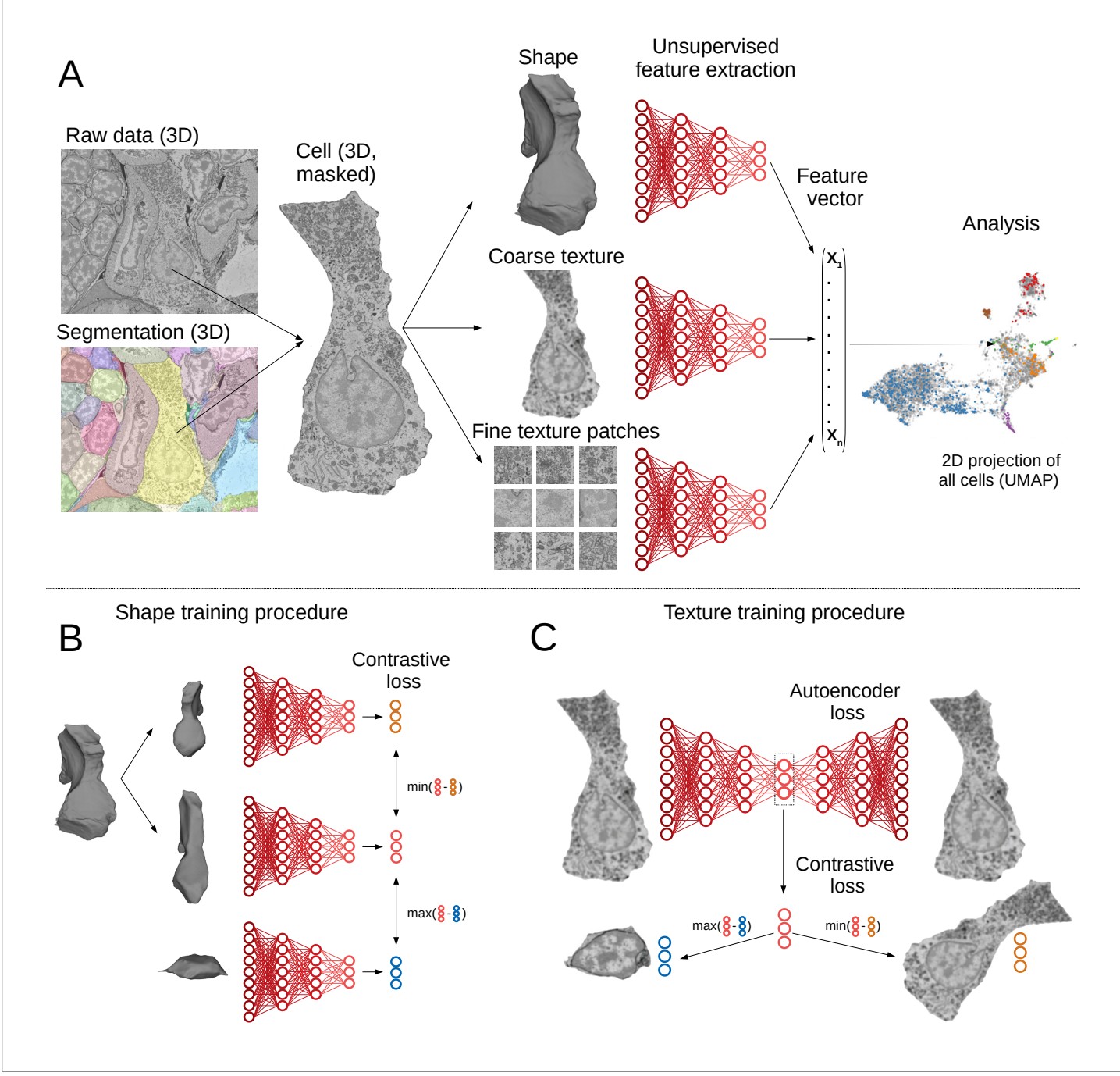

**Figure 1.** Deep-learning pipeline for extracting MorphoFeatures. (**A**) Cell segmentation is used to mask the volume of a specific cell (and its nucleus) in the raw data. Neural networks are trained to represent shape, coarse, and fine texture from the cell volume (separately for cytoplasm and nuclei). The resulting features are combined in one MorphoFeatures vector that is used for the subsequent analysis. (**B**) Training procedure for the shape features. A contrastive loss is used to decrease the distance between the feature vectors of two augmented views of the same cell and increase the distance to another cell. (**C**) Training procedure for the coarse and fine texture features (here illustrated by coarse texture). Besides the contrastive loss, an autoencoder loss is used that drives the network to reconstruct the original cell from the feature vector.

maximal amount of information about the cells, while enforcing distances between feature vectors to reflect morphological similarity of cells.

The pipeline was trained and applied on the cellular atlas of the marine annelid *P. dumerilii* (*Vergara et al., 2021*). It comprises a 3D SBEM volume of the whole animal that has sufficient resolution to

distinguish ultrastructural elements (organelles and inclusions, nuclear and cytoplasm texture, etc.) and an automated segmentation of 11,402 cells and nuclei (*Figure 1A*). Additionally, whole-animal gene expression maps are available that cover many differentiation genes and transcription factors.

Such a high number of morphologically and genetically diverse cells can make the initial exploratory analysis difficult. We designed MorphoFeatures to enable unbiased exploration of morphological variability in the animal on both cellular and tissue level; and indeed our training procedure yields 480 features that extensively describe each cell in terms of its cytoplasm and nucleus shape, coarse texture, and fine texture (80 features for each category). The features can be found at https://github.com/kreshuklab/MorphoFeatures.git; the feature table can be directly integrated with the *Platynereis*

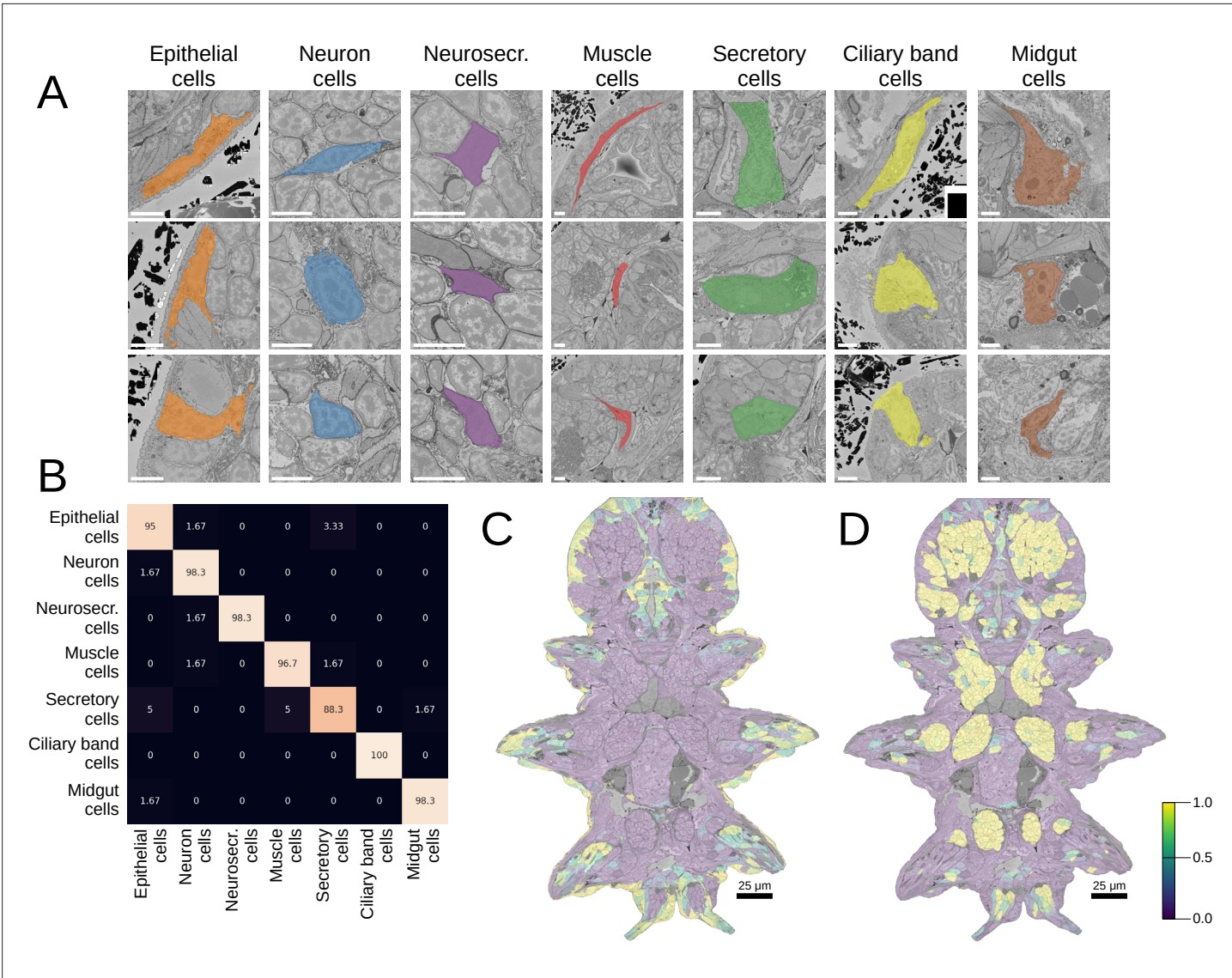

**Figure 2.** Morphological class prediction. (**A**) Examples of cells from seven manually defined morphological classes used for evaluation. (**B**) Confusion matrix of class prediction for the logistic regression model. Rows are the labels, and columns are the predictions. Scale bars: 5 µm. (**C**) The predicted probability of the epithelial class in the whole animal. Note that while a few cells on the animal surface have been used for training of the logistic regression, no labels were given in the foregut opening or chaetae which are still correctly recognised as epithelial. (**D**) The predicted probability of neural cells in the whole animal.

The online version of this article includes the following figure supplement(s) for figure 2:

**Figure supplement 1.** Morphological class predictions.

**Figure supplement 2.** Confusion matrix of class predictions for the logistic regression model using separate morphological components.

atlas of *Vergara et al., 2021* through the MoBIE plugin (*Pape et al., 2022*) in Fiji (*Schindelin et al., 2012*).

## MorphoFeatures allow for accurate morphological class prediction

Good morphological features should distinguish visibly separate cell groups present in the data. To estimate the representation quality of our MorphoFeatures, we quantified how well they can be used to tell such groups apart. We took the morphological cell class labels available in the dataset, proofread, corrected, and expanded them. These labels include seven general classes of cells: neurons, epithelial cells, midgut cells, muscle cells, secretory cells, dark neurosecretory cells, and ciliary band cells (*Figure 2A*) with ~60 annotated cells per class in the whole volume. It is worth noting that these cell classes were chosen based on visual morphological traits and do not necessarily represent genetically defined cell type families (*Arendt et al., 2019*). For example, while ciliary band cells are a small narrow group of morphologically homogeneous cells that likely constitute one cell type family, secretory cells were defined as cells with abundant organelles/inclusions in their cytoplasm that could indicate secretion, spanning a wider range of morphologies and, potentially, genetically defined cell types.

We extracted MorphoFeatures by passing all the cells through the neural network trained as described above (*Figure 1A*) and used them as an input to train a logistic regression classifier to predict the above-mentioned cell classes. This classifier has few parameters and can be trained even on the modest number of labels we have available. If any subset of the features strongly correlates with a given cell class, the model will show high performance. A classifier with MorphoFeatures achieved an accuracy of 96%, showing that the learned features are sufficient to distinguish broad morphological classes. To better understand remaining inaccuracies, we show the errors made by the classifier in *Figure 2B* and *Figure 2—figure supplement 1A*. This reveals, as expected, that while morphologically homogeneous classes can be easily separated, the classifier tends to confuse some secretory cells with other types (midgut and epithelial cells) that are also difficult to discern by eye and might well be mis-annotated. It also confuses the general neuron class with neurosecretory cells that represent a subclass of neurons.

To illustrate large-scale performance of the broad cell type classification model, we show its predictions on the whole animal in *Figure 2C and D* and *Figure 2—figure supplement 1B*. These results demonstrate that MorphoFeatures are sufficiently expressive to enable training a generalisable model from a small number of labels, taking epithelial cells as a showcase: while the model was trained exclusively on the cells of the outer animal surface, it successfully predicted the cells outlining the gut opening and the chaetae cells as epithelial as well. Even more convincingly, the model predicted some cells in the midgut region as neurons, which after careful visual examination were indeed confirmed to be enteric neurons.

To explore to which extent each of the morphology components (e.g. cell shape or fine nuclear texture) contributes to predicting the manually labelled cell classes, we ran a similar classification pipeline using these components separately (*Figure 2—figure supplement 2*). The results show that, for example, cytoplasm texture is sufficient to tell some classes apart with high precision. However, both coarse and fine cytoplasm texture perform slightly worse on distinguishing neurosecretory cells, in which nuclei occupy almost the whole cell volume. Cell shape is satisfactory to characterise neurons and muscle cells but shows inferior performance on epithelial and secretory cells, which display a great variety of cell shapes. Surprisingly, nuclear fine texture features correctly distinguish, among other classes, muscle cells, suggesting that this class has characteristic chromatin texture.

In *Vergara et al., 2021*, morphological classification of cells was performed based on manually defined features, including descriptors of shape (e.g. volume, sphericity, and major and minor axes), intensity (e.g. intensity mean, median and SD), and texture (Haralick features) extracted from the cell, nucleus, and chromatin segmentations, with 140 features in total. To compare these explicitly defined features with implicitly learned MorphoFeatures, we passed them through the same evaluation pipeline yielding a classifier which achieves the accuracy of 94%, in comparison to 96% achieved by MorphoFeatures. The classifier mostly made similar mistakes but performed worse on the classes of muscle and epithelial cells. Both sets of features demonstrate high accuracy on the broad morphological type classification task. For a more detailed comparison, we turn to a different task which allows us to implicitly, but quantitatively, evaluate how well the features group similar cells together.

We use the metric introduced in *Vergara et al., 2021* that is based on the fact that the animal is bilaterally symmetric, thus almost all the cells on one side of the animal will have a symmetric partner. The metric estimates how close a given cell is to its potential symmetric partner in the morphological feature space in comparison to all the other cells in the animal (see Materials and methods). According to this metric, our MorphoFeatures are 38% more precise in locating a symmetric partner than the ones from *Vergara et al., 2021*, showing the widely accepted superiority of neural network extracted features to explicitly defined ones.

## MorphoFeatures uncover groups of morphologically similar cells

Comprehensive analysis of morphological features should enable the grouping of cells with similar morphology, which may represent cell types. We investigated the grouping induced by Morpho-Features by projecting the features of all the animal cells onto 2D space using the dimensionality reduction technique UMAP (*McInnes et al., 2018*). To explore the quality of our representations, we repeatedly took a random cell from the animal and visualised its three closest neighbours (*Figure 3A*). This shows that cells in close proximity in the feature space appear highly visually similar and might represent cell types at the morphological level. For example, a group of secretory cells in which the cytoplasm is filled with round electron-transparent vesicles (upper green panel) corresponds to the 'bright droplets' cells from *Verasztó et al., 2020*. Another group of secretory cells has an extensive endoplasmic reticulum (ER) surrounding the nucleus with a prominent nucleolus (lower green panel) and represents parapodial spinning gland cells (*Verasztó et al., 2020*). One can also see clear groups of flat developing midgut cells (brown panel), short muscles (red panel), ciliary band cells on the outer animal surface (yellow panel) as well as extended flat and short round epithelial cells (orange panels). Another interesting observation is that even though the neurite morphology is not taken into account, neurons show remarkable diversity based on soma morphology only (blue panels). Careful examination of a group of cells, initially labelled as neurons, that are located next to the epithelial cells on the UMAP projection (upper right blue panel) revealed that these cells are sensory cells that extend their processes to the epithelial surface.

To explore the structure of the MorphoFeatures space, we performed hierarchical clustering of all cells. In the first clustering round, we split cells into 15 broad morphological groups (*Figure 3B*). Cluster content could be revealed through available annotations and visualisation with the Platy-Browser (*Pape et al., 2022*; *Figure 3B*): Clusters 1–8 represent different groups of neurons; 9–11 epithelial cells; 12 and 13 are muscles; 14 midgut cells; while cluster 15 was further subclustered into secretory cells (subcluster 1) and ciliary band cells (subcluster 2). MorphoFeatures can thus guide the morphological exploration of a whole-animal volume, facilitating the discovery and annotation of morphologically coherent groups of cells (*Figure 3B*). For example, the foregut epithelial cells are distinct from the epithelial cells of the outer animal surface and chaetae cells (clusters 9, 11, and 10, respectively). Separate groups are also formed by different types of neurons, for example, foregut neurons (cluster 4).

## Morphological clusters have distinct gene expression profiles

To further describe the morphological clusters, we took advantage of the whole-animal cellular gene expression atlas available for *Platynereis* (*Vergara et al., 2021*), containing more than 200 genes mapped onto the EM volume. We looked for genes that are highly expressed in a given cluster, while also being specific for this cluster - having low expression in the other cells of the animal.

Many of our clusters show a clear genetic signature (*Figure 4A*, *Figure 4—figure supplement 1*). Among the neurons, cluster 5 shows the most specific gene expression including the Transient receptor potential (Trp) channels *pkd2*, *trpV4*, and *trpV5*, and the bHLH transcription factor *asci*, which demarcate sensory cells of the palpal, antennal, and cirral ganglia (*Vergara et al., 2021*). Cluster 1 shows specific expression of the homeodomain transcription factors *lhx6* and *phox2* and an *ap2* family member, while cluster 6 composed of dark neurosecretory cells shows specific expression of two neurosecretion markers, atrial natriuretic peptide receptor *anpra* and prohormone convertase *phc2*, identifying these as brain parts of the circular and apical nervous system, respectively (*Vergara et al., 2021*; *Arendt, 2021*). Cluster 8, another neural cluster, is enriched for the oxidative stress marker *cytoglobin* (*globin-like*; *Song et al., 2020*) and the bHLH transcription factor *mitf*. Subclustering revealed three subclusters, one of which represents rhabdomeric photoreceptors of the adult

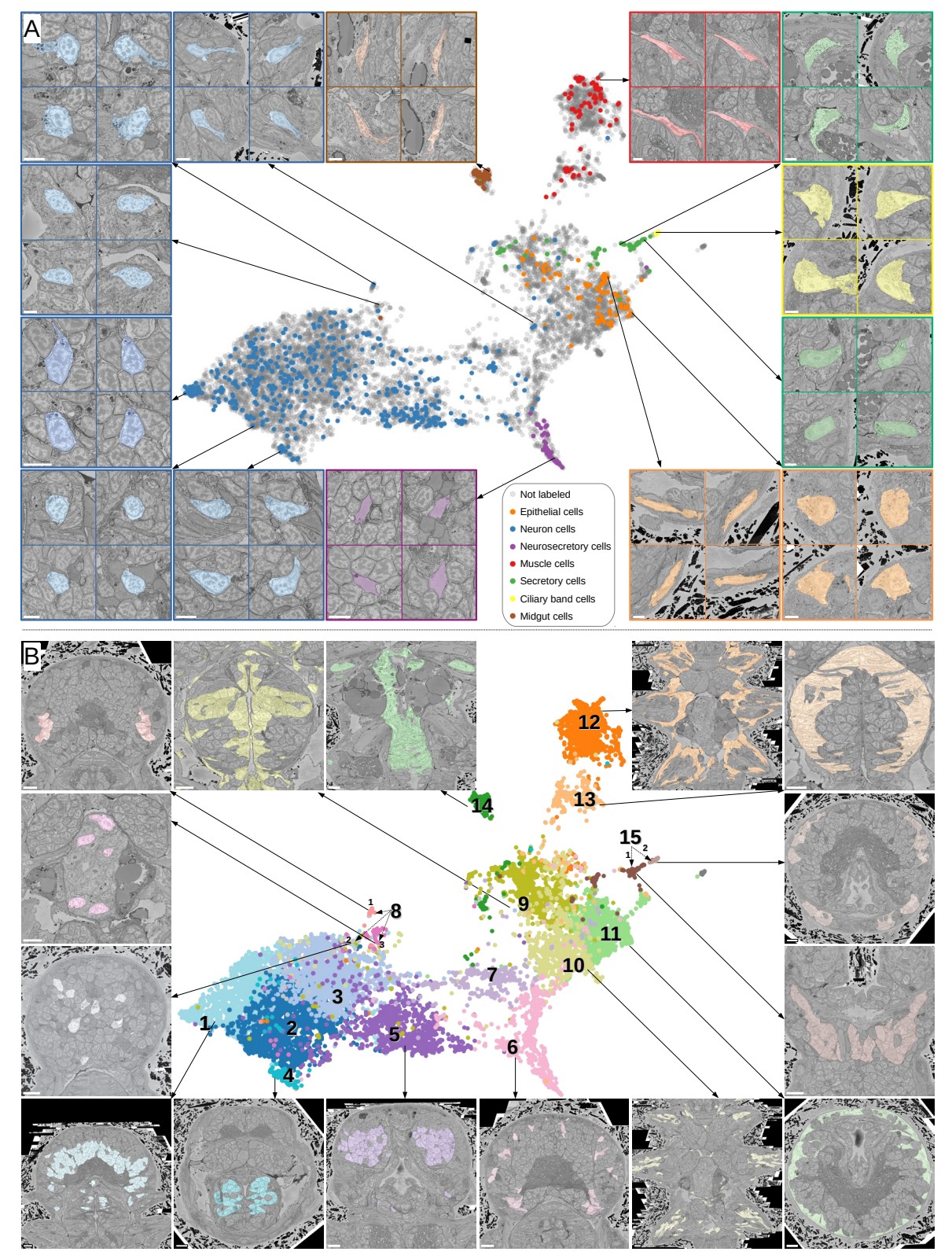

**Figure 3.** Visual analysis of MorphoFeatures representations. (**A**) Finding visually similar cells using MorphoFeatures. Multidimensional features are visualised in 2D using UMAP. Each point represents a feature vector of a cell from the dataset. The cells for which annotations are available are visualised in respective colours. For a random cell, the cell and its three closest neighbours in the UMAP space are visualised in the electron microscopy (EM) volume. Scale bars: 5 μm. (**B**) Visualising morphological clusters. Clustering results are visualised on the MorphoFeatures UMAP representation. For

*Figure 3 continued on next page*

*Figure 3 continued*

some clusters, the cells comprising the cluster are shown in the animal volume to visualise the cell type. For example, cluster 6 precisely picks out the dark neurosecretory cells, while cluster 14 corresponds to the midgut cells (see Text for more details). Scale bars: 10µm.

The online version of this article includes the following figure supplement(s) for figure 3:

**Figure supplement 1.** Cluster of split segmentation errors.

**Figure supplement 2.** Cells with segmentation errors.

eye (*Figure 3B*, subcluster 8.1) that specifically express *mitf* and *globin-like*. An average cell in this subcluster (*Figure 4B*) shows abundant black inclusions. The second subcluster (8.2) contains pigment cells with black pigment granules. Cells in the third subcluster (8.3) are located in the midgut, with a smooth oval shape, neurite extensions and nuclei taking up the most cell volume (*Figure 4C and D*) - in striking contrast to other midgut cells but resembling neurons, making it likely that these cells

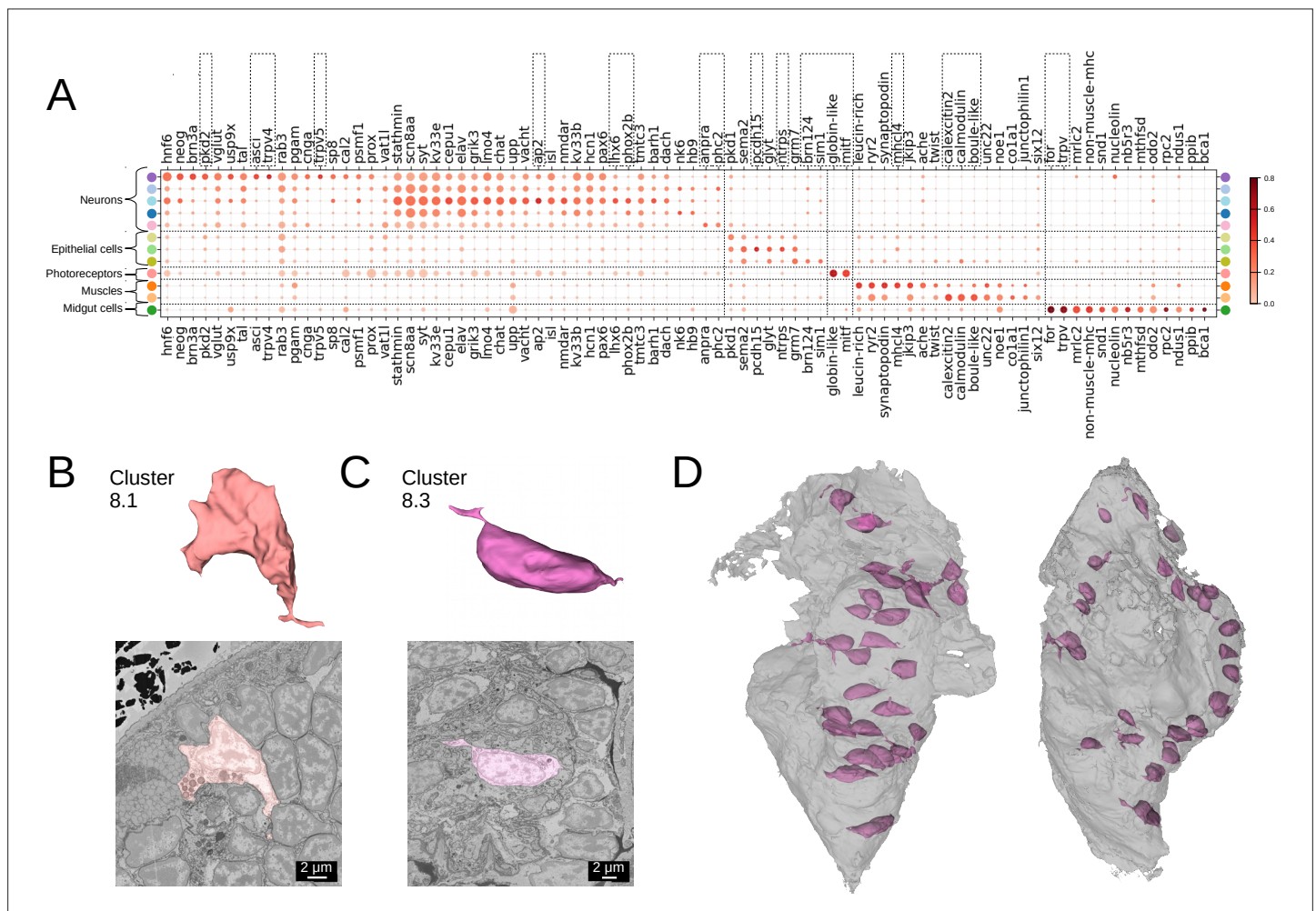

**Figure 4.** Clustering and gene analysis. (**A**) Gene expression dot plot. The size of the dots shows how much of a cluster expresses a gene; the colour shows how much of the expression of a gene is confined to a cluster (see Materials and methods). The genes mentioned in the text are enboxed. The clusters lacking highly specific gene expression were not included. (**B-C**) The average shape and texture (see Materials and methods) of (**B**) rhabdomeric photoreceptors (cluster 8.1) and (**C**) the enteric neurons (cluster 8.3). (**D**) Localisation of the enteric neurons (pink) in the midgut volume (grey). Left: frontal view and right: side view.

The online version of this article includes the following figure supplement(s) for figure 4:

**Figure supplement 1.** Clustering and gene analysis.

**Figure supplement 2.** Comparing MorphoFeatures to a set of manually defined features from *Vergara et al., 2021*.

represent enteric neurons previously postulated on the basis of serotonin staining (*Brunet et al., 2016*).

Among the epidermal cells, the outer cluster 11 is enriched for *protocadherin* (*pcdh15*) and *neurotrypsin* (*ntrps*), whereas cluster 9 representing foregut epidermis shows high specificity for the bHLH transcription factor *sim1* and the homeodomain factor *pou3* (*brn124*, known to be expressed in stomodeal ectoderm in sea urchin; *Cole and Arnone, 2009*). For the muscles, the specific expression of striated muscle-specific myosin heavy chain *mhcl4* in cluster 12 denotes the group of striated somatic muscles, while its absence in cluster 13 indicates that the foregut muscles have not yet switched to expressing striated markers (*Brunet et al., 2016*). Instead, the high and specific expression of the calexcitin-like sarcoplasmic calcium-binding protein *Scp2* (*calexcitin2*; *White et al., 2011*), *calmodulin*, and of the Boule homologue *Boll* (*boule-like*) in cluster 13 suggests a calcium excitation and/or sequestration mechanism specific for foregut muscles. Finally, the cluster of midgut cells (cluster 14) specifically expresses the forkhead domain transcription factor *foxA* (*for*), a gut developmental marker (*Boyle and Seaver, 2008*) and *TrpV-c*, another Trp channel paralogue of the vanilloid subtype with presumed mechanosensory function.

Beyond that, we noted considerable genetic heterogeneity in the midgut cluster. Finer clustering (*Figure 5*) revealed one subcluster with strong expression of the smooth muscle markers *non-muscle-mhc* and *mrlc2*, and another expressing a chitinase related to chitotriosidase (*nov2*) and the zinc finger transcription factor *Prdm16/mecom*, known to maintain homeostasis in intestinal epithelia (*Stine et al., 2019*). Visualising the cells belonging to each subcluster and the specifically expressed genes in the PlatyBrowser (*Figure 5E*) revealed distinct territories in the differentiating midgut, which we interpret as midgut smooth musculature and digestive epithelia. We also detected an enigmatic third cluster located outside of the midgut, in the animal parapodia, comprising cells that resemble midgut cells morphologically (*Figure 5D*). For this subcluster, the current gene repertoire of the cellular expression atlas did not reveal any specifically expressed gene.

## MorphoFeatures correspond to visually interpretable morphological properties

Neural networks are often referred to as 'black boxes' to signify that it is not straightforward to trace learned features or decisions back to particular input properties. To examine whether it is possible to understand which properties of cells learned by our network distinguish the discovered morphological groups, we first identified MorphoFeatures that have high values specific to each cluster (*Figure 6—figure supplement 1*). Then for a set of characteristic features, we visualised cells that correspond to the maximum and minimum value of the corresponding feature (*Figure 6*). Visual inspection showed that many of them can be matched to visually comprehensible properties.

For example, cytoplasm coarse texture feature 21 (*Figure 6*, upper left), which distinguishes the outer epithelial cells (cluster 11), shows its minimal value in secretory cells, which contain multiple highly distinct types of texture, and its maximal value in the cells crowded with small vesicles and cisternae on the surface of the animal. The cluster of enteric neurons (cluster 8.3) strongly differs from other neurons by nuclear coarse texture feature 4 (*Figure 6*, upper right), which contrasts nuclei with large amount of heterochromatin often attached to the nuclear lamina to nuclei with a high amount of euchromatin and prominent nucleoli. Cytoplasm fine texture feature 50 (*Figure 6*, middle left), characteristic to the midgut cells (cluster 14), is the lowest in cells with a small stretch of homogeneous cytoplasm with mitochondria and has its peak in cells with abundant Golgi cisternae and medium-sized mitochondria. Rhabdomeric photoreceptors of the adult eye (cluster 8.1) display specific nuclear fine texture feature 7 (*Figure 6*, middle right) that differentiates between nuclei with smooth and rather grainy texture. Cell shape feature 14 (*Figure 6*, lower left) has its minimal value found in beaker-shaped cells with smooth surface and its maximal value in rather round cells with extremely rugged surface and is specific for ciliary band cells (cluster 15.2). Foregut muscles (cluster 13) can be described using nuclear shape feature 66 (*Figure 6*, lower right), having one extreme in nuclei with a small compartment with a rough surface (as a result of a segmentation error) and the other extreme in elongated flat nuclei.

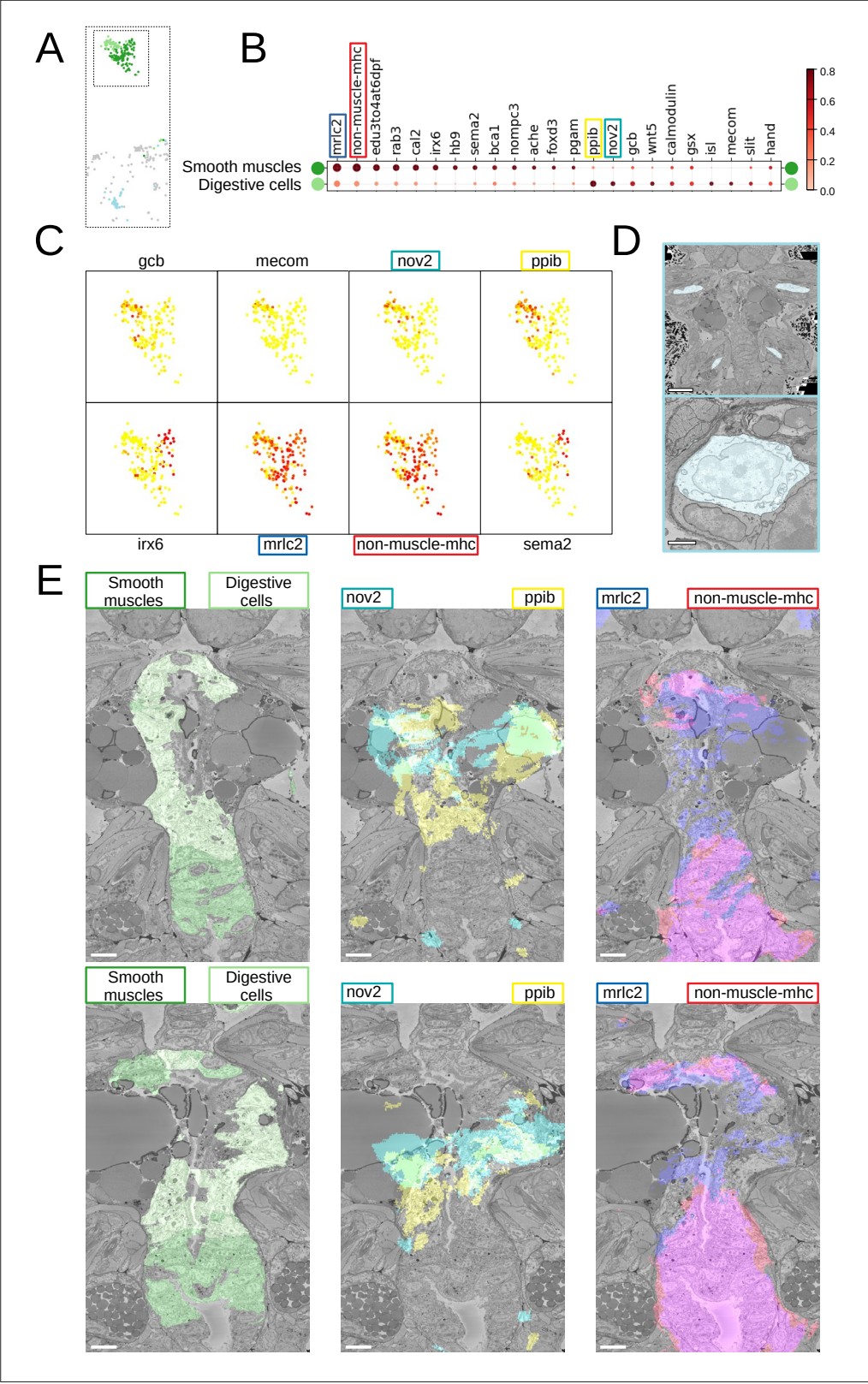

**Figure 5.** Midgut cell types with defining genes. (**A**) Finer clustering of the midgut cluster results in three subclusters. (**B**) Gene expression dot plot of the two midgut subclusters that are presumably developing smooth muscles and digestive cells of the midgut. The size of the dots shows how much of a cluster expresses a gene; the colour shows how much of the expression of a gene is confined to a cluster (see Materials and methods). (**C**) Some

*Figure 5 continued on next page*

*Figure 5 continued*

of the genes shown to be differentially expressed in the two subclusters, plotted on the UMAP representation. (**D**) The location (upper panel) and an example cell (lower panel) of the subcluster located in the animal parapodia. Scale bars: upper panel - 25µm, lower panel - 5µm. (**E**) Cells belonging to the two cell types (left panels) and the genes differentiating them (centre and right panels) are visualised in the animal volume, with colour representing gene expression overlayed on the electron microscopy (EM) plane view. Scale bar: 10µm.

## Adding neighbour morphological information helps to identify tissues and organs

Most cells do not function in isolation but rather form tissues, i.e., groups of structurally and functionally similar cells with interspersed extracellular matrix. We set out to systematically identify tissues with a new feature vector assigned to each cell that takes into account information about the morphology of neighbouring cells. For this, we combined the MorphoFeatures of a cell with the average MorphoFeatures of its immediate neighbours, yielding a feature vector that represents the morphology of both the cell and its surrounding, which we refer to as MorphoContextFeatures.

Applying the representation analysis described before, we find that proximity in MorphoContextFeature space no longer reflects morphological similarity only but rather identifies neighbouring groups of morphologically similar cells (whereby the neighbouring groups can be morphologically dissimilar). In other words, we now identify different tissues that together constitute organs composed of various tissues (*Figure 7*). For example, a separate assembly of groups of cells in the lower part of the UMAP projection represents the animal foregut subdivided into its constituting tissues comprising foregut neurons (lower blue panel), foregut muscles (lower beige panel), and foregut epithelium (left yellow-green panel). Next to this, we identify a group of muscles surrounding the foregut (left red panel).

For the nervous system, adding neighbourhood information helps distinguishing cirral, palpal, and dorsal ganglia (three upper right panels, from left to right), as well as the ventral nerve cord and peripheral ganglia (two lower right panels). To benchmark the quality of this grouping, we clustered all the cells in the MorphoContextFeature space (see Materials and methods) and compared the neuron clusters to both manually segmented ganglionic nuclei and to the genetically defined neuronal clusters taking into account available gene expression in the cellular atlas (*Vergara et al., 2021*; *Figure 8A and B*). Notably, manual segmentation of brain tissues relied on visible tissue boundaries and thus has lower quality in the areas where such boundaries are not sufficiently distinct.

In essence, all three ways of defining ganglia lead to very similar results (*Figure 8B*), yet MorphoContextFeature clustering appeared to be the most powerful. Gene clustering failed to distinguish the circumpalpal ganglion, defined both by manual segmentation and MorphoContextFeature clustering, due to the lack of distinct gene expression in the atlas. Manual segmentation failed to distinguish between the dorso-posterior and dorsal-anterior ganglionic nuclei due to the lack of distinct tissue boundaries. These are well defined by gene clustering and subdivided even further by MorphoContextFeature clustering. Unaware of the physical tissue boundaries, MorphoContextFeatures sometimes led to ambiguous assignment of cells neighbouring other tissues, especially cells with strongly different morphology. Such noisy assignment can be noticed, for example, in the neurons bordering muscle cells on the boundary of the cirral ganglia and in the centre of the brain.

To further examine the contribution of incorporating cellular neighbourhood morphology, we visualise the discovered foregut tissues and ganglia on the MorphoFeatures representation (*Figure 8—figure supplement 1* and *Figure 9—figure supplement 1*). This shows that while some of the tissues, such as palpal ganglia or foregut epithelium, are composed of cells of one morphological type, others, e.g., circumpalpal ganglia or infracerebral gland, comprise cells with different morphology and could only be detected when taking into account cellular context as well.

## Unbiased exploration of foregut tissues

To further illustrate how MorphoContextFeatures can assist an unbiased exploration of an animal EM volume, we focused on the foregut. Subclustering whenever suitable (*Figure 9A*; see Materials and methods), the resulting clusters revealed various foregut tissues including foregut epidermis, ganglia, inner musculature, and a prominent muscle surrounding the foregut. We also found a tissue of secretory neuron-like cells that surround the foregut like a collar (*Figure 9B and C*, in green).

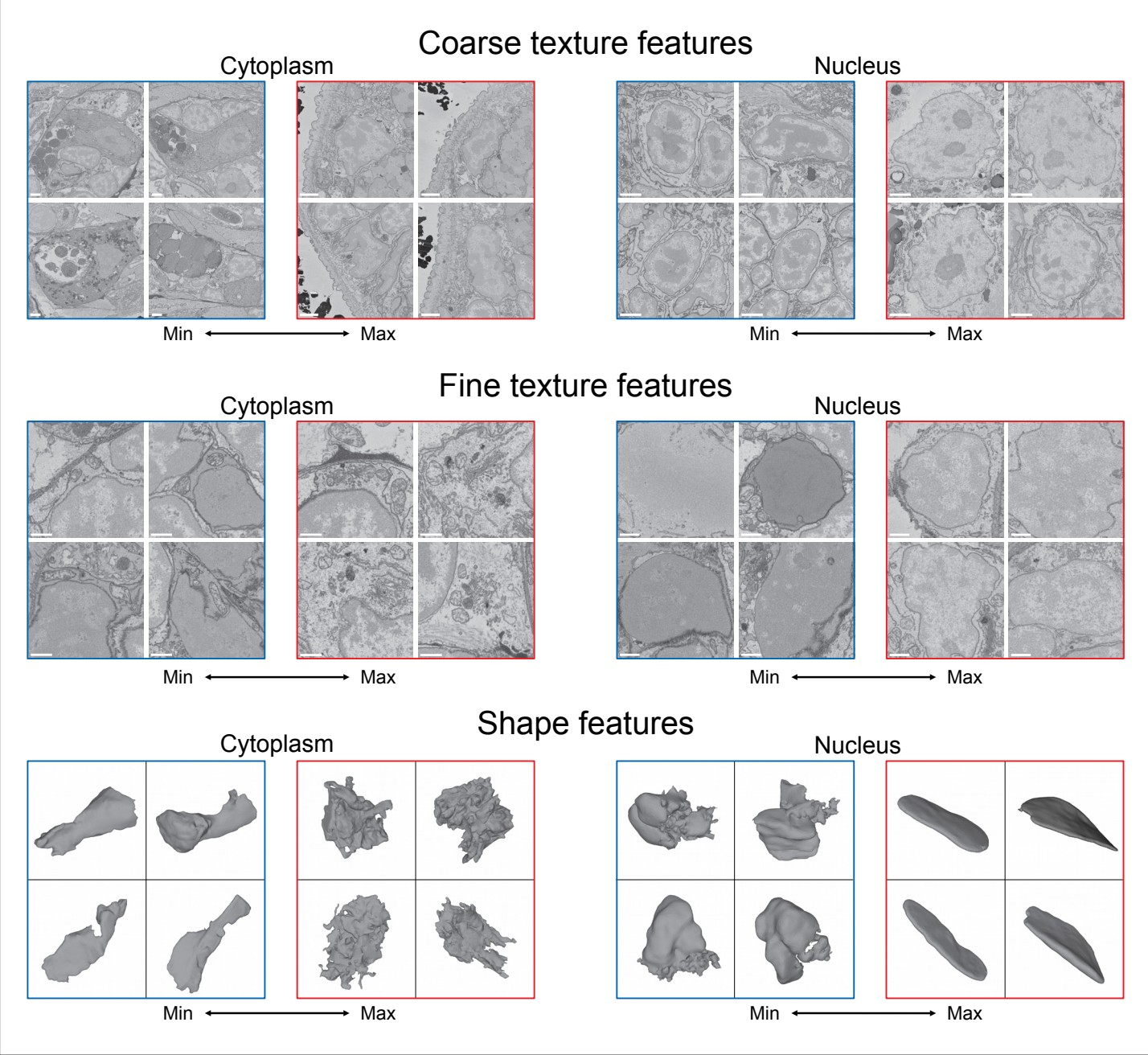

**Figure 6.** Visualisation of the learned features. For each feature, four cells with a minimal (blue) and four cells with a maximal (red) value of the feature are shown, see text for detailed analysis. Shown are cytoplasm coarse texture feature 21, nuclear coarse texture feature 4, cytoplasm fine texture feature 50, nuclear fine texture feature 7, cytoplasm shape feature 14, and nuclear shape feature 66. Scale bars: coarse texture - 2µm, fine texture - 1µm.

The online version of this article includes the following figure supplement(s) for figure 6:

**Figure supplement 1.** Dot plot of MorphoFeatures specific to clusters (**A**) 10–14, 8.1, 8.2, and 3, (**B**) 1, 4, 8.3, 15.1, and 7, and (**C**) 6 and 15.2.

Further inspection revealed that the latter structure is located underneath the brain neuropil and close to the neurosecretory endings of the neurosecretory plexus (*Verasztó et al., 2020*) and is bounded by thin layers of longitudinal muscle fibres on both the outer and the inner surface (*Figure 9—figure supplement 2A*). The location of the structure indicated it might be the differentiating infracerebral gland - a neurosecretory gland located beneath the brain assumed to play a role in sensing blood glucose levels (*Backfisch, 2013*; *Baskin, 1974*; *Hofmann, 1976*; *Golding, 1970*). The organ is leaf-shaped and lays between the posterior pair of adult eyes (*Figure 9—figure*

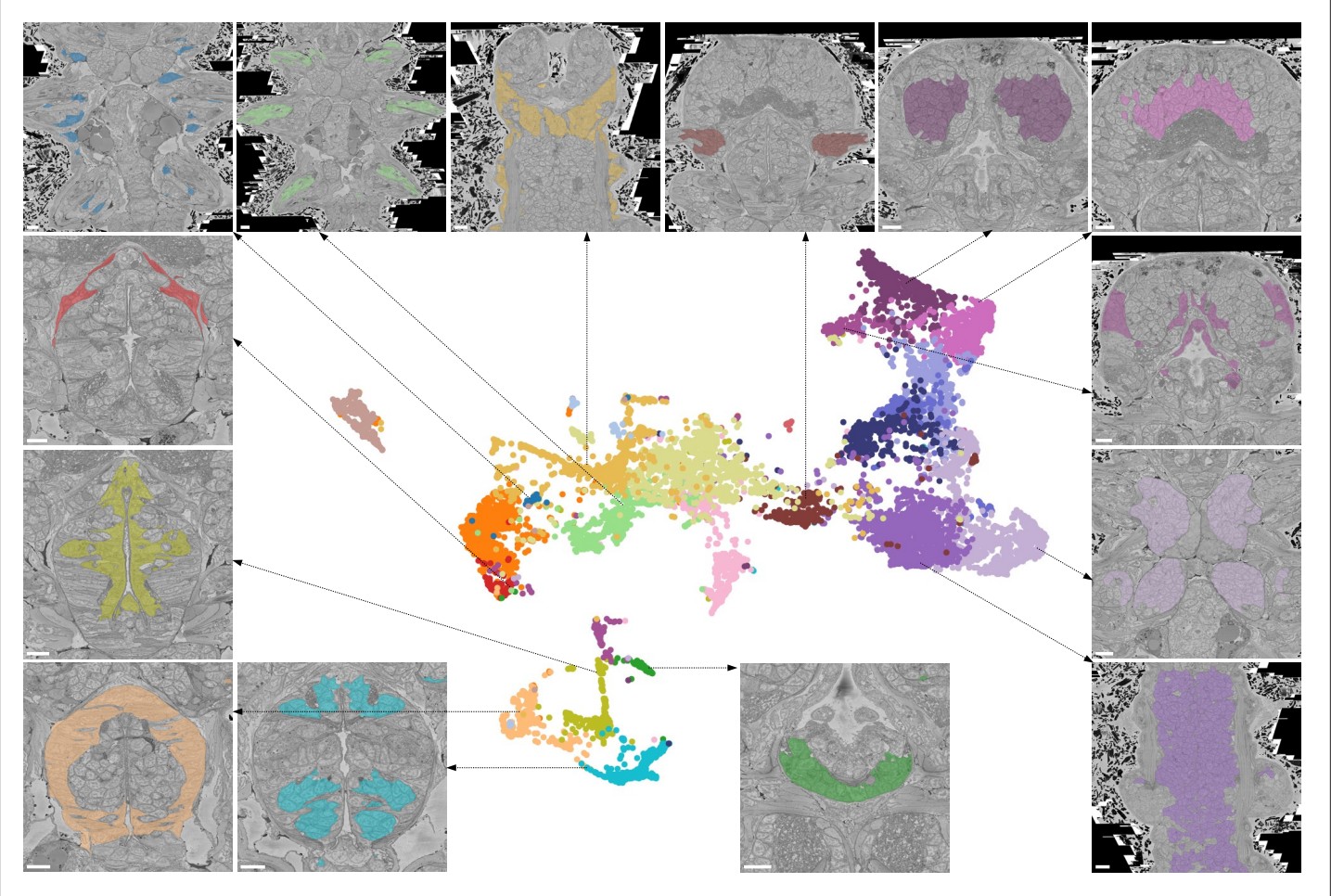

**Figure 7.** Characterising neighbourhoods with MorphoContextFeatures. Clustering results are visualised on the MorphoContextFeatures UMAP representation. For some clusters, the cells comprising the cluster are shown in the animal volume to visualise the cell type. Upper panels (from left to right): secretory (blue) and epithelial (green) cells of parapodia, epithelial, and secretory cells of the head, cirral, palpal, and dorsal ganglia. Lower panels (from left to right): foregut muscles, foregut neurons, infracerebral gland, and ventral nerve cord. Left panels (from top to bottom): muscles surrounding the foregut and foregut epithelium. Right panels (from top to bottom): epithelial-sensory circumpalpal cells and peripheral ganglia. Scale bars: 10μm.

supplement 2B) as also described in *Backfisch, 2013* and borders a cavity likely to be a developing blood vessel (*Figure 9—figure supplement 2C*). In comparison to the foregut neurons that express sodium channel *scn8aa*, the infracerebral gland cells specifically express homeobox protein *onecut/ hnf6* (*Figure 9D and E*). We also noted that the tissue stains positive for EdU applied between 3 and 5 dpf, indicating that it is still proliferating.

We then identified the prominent muscles surrounding the foregut as the anterior extension of the axochord (*Lauri et al., 2014*). Visualising the genes specifically expressed in these muscles, we noticed specific expression of the type I collagen gene *col1a1*, a marker for axochordal tissue, and of muscle junctional gene *junctophilin1* (*Figure 9F*). An anterior extension of the axochord around the foregut has been observed in several species and is described in *Lauri et al., 2014*; *Brunet et al., 2016*; *Nielsen et al., 2018*.

## Discussion

We presented an automated method of extracting comprehensive representations of cellular morphology from segmented whole-animal EM data. In stark contrast to the currently available methods, ours refrains both from specifying features explicitly and from predefining a concrete task that might skew the method towards extracting specific features: putting emphasis on some features

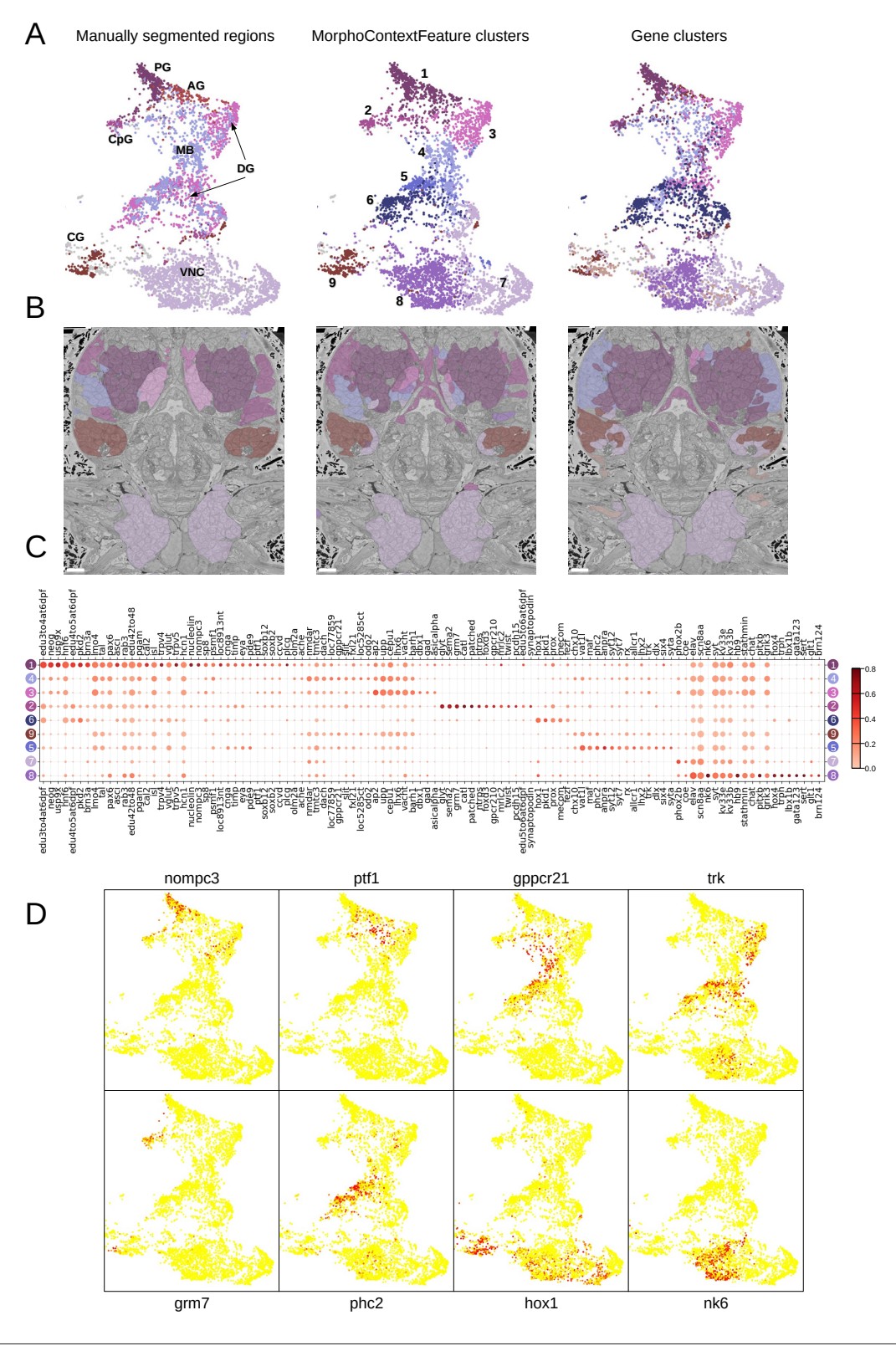

**Figure 8.** MorphoContextFeatures define ganglionic nuclei. (**A and B**) The animal ganglia as defined by manual segmentation (**Vergara et al., 2021**), our MorphoContextFeature clustering and gene expression clustering displayed on (**A**) the UMAP representation and (**B**) in the animal volume. PG, palpal ganglia; CpG, circumpalpal ganglia; AG, antennal ganglia; MB, mushroom bodies; DG, dorsal ganglion; CG, cirral ganglia; VNC, ventral nerve

*Figure 8 continued on next page*

*Figure 8 continued*

cord. Scale bars: 10μm. (**C**) Gene expression dot plot of the ganglia defined by MorphoContextFeature clustering. The size of the dots shows how much of a cluster expresses a gene; the colour shows how much of the expression of a gene is confined to a cluster (see Materials and methods). (**D**) Some of the genes shown to be differentially expressed in the ganglia defined by MorphoContextFeature clustering, plotted on the UMAP representation.

The online version of this article includes the following figure supplement(s) for figure 8:

**Figure supplement 1.** Ganglia clusters on the MorphoFeatures representation.

may bias descriptions towards specific morphologies or cell types, hindering exploratory analysis. To obtain extensive generally useful representations, we rely on the latest progress in self-supervised deep learning and set only two criteria for the extracted features: (1) they have to be sufficiently detailed to allow reconstructing the original cell volume and (2) they have to bring visually similar cells in feature space closer to each other than dissimilar ones. We train a deep-learning pipeline optimised for these two conditions and show that it produces rich cellular representations, combining shape-, ultrastructure-, and texture-based features. More precisely, we show that the obtained representations (MorphoFeatures) capture fine morphological peculiarities and group together cells with similar visual appearance and localisation in the animal. Clustering of cells in the MorphoFeature space and taking advantage of an existing gene expression atlas with cellular resolution, we show that the obtained clusters represent units with differential and coherent gene expression.

Both for morphological type and for tissue detection, we illustrate how MorphoFeatures can facilitate unbiased exploratory analysis aimed at detecting and characterising unusual phenotypes or unusual cell neighbourhoods. Moreover, we show quantitative superiority of MorphoFeatures over a broad range of manually defined features on the tasks of cell classification and symmetric partner detection. We also show that, despite being learned implicitly, MorphoFeatures still correspond to visually understandable properties and can often be interpreted. Finally, adding morphological information from the neighbouring cells (MorphoContextFeatures), we reveal the power of our approach to detect local groups of cells with similar morphology that represent tissues.

Since our pipeline is based on the fast-evolving field of self-supervised deep learning, we expect it to further improve with its general advancement. For example, a promising direction are models, such as *Grill et al., 2020*, that only rely on positive samples and remove the need to define which cells belong to a different type. Besides MorphoFeatures, we also believe that MorphoContextFeatures, which incorporate neighbourhood information, will potentially benefit from state-of-the-art self-supervised techniques such as contrastive-learning techniques on graphs (*Hassani and Khasahmadi, 2020*; *Velickovic et al., 2019*). Additionally, we expect hardware improvement, i.e., GPU RAM size, to allow training bigger models on larger texture patches of higher resolution that might further refine the quality of morphological representations. Another interesting direction enabled by the modularity of our pipeline is to integrate data from other modalities. Our features currently integrate shape and texture information of cells, but one could also envision enriching cellular descriptions by adding different types of data, e.g., transcriptomics, proteomics, or metabolomics, in a similar manner.

## Versatility of the MorphoFeatures toolbox

We envision our pipeline to enable, for the first time, fast systematic investigation of large EM volumes that involves consistent grouping and characterisation of morphological types, discovering particular phenotypes and neighbourhoods, as well as localising cells or tissues in an animal volume that are visually similar to cells or groups of cells of interest. Moreover, we expect MorphoFeatures to be useful for studies where several image volumes of the same species are acquired, be it another sample, a varying phenotype, or a different developmental stage. In such cases, training the pipeline on multiple volumes simultaneously to produce sample-independent features would result in morphological representations that can be used to map cells to their corresponding partners in another volume, enabling unbiased comparative analysis at scale. In addition, in case the other volumes are not segmented, one could use the fine texture part of MorphoFeatures to locate patches with phenotypic characteristic signature or cells with distinct ultrastructure (e.g. muscles and secretory cells). Using fine texture descriptors in such cases can also facilitate further segmentation since the segmentation of the previous volume can be used to define constraints of which textures can co-occur in the cells of the animal. However, given the high sensitivity of neural networks to intensity variations, it

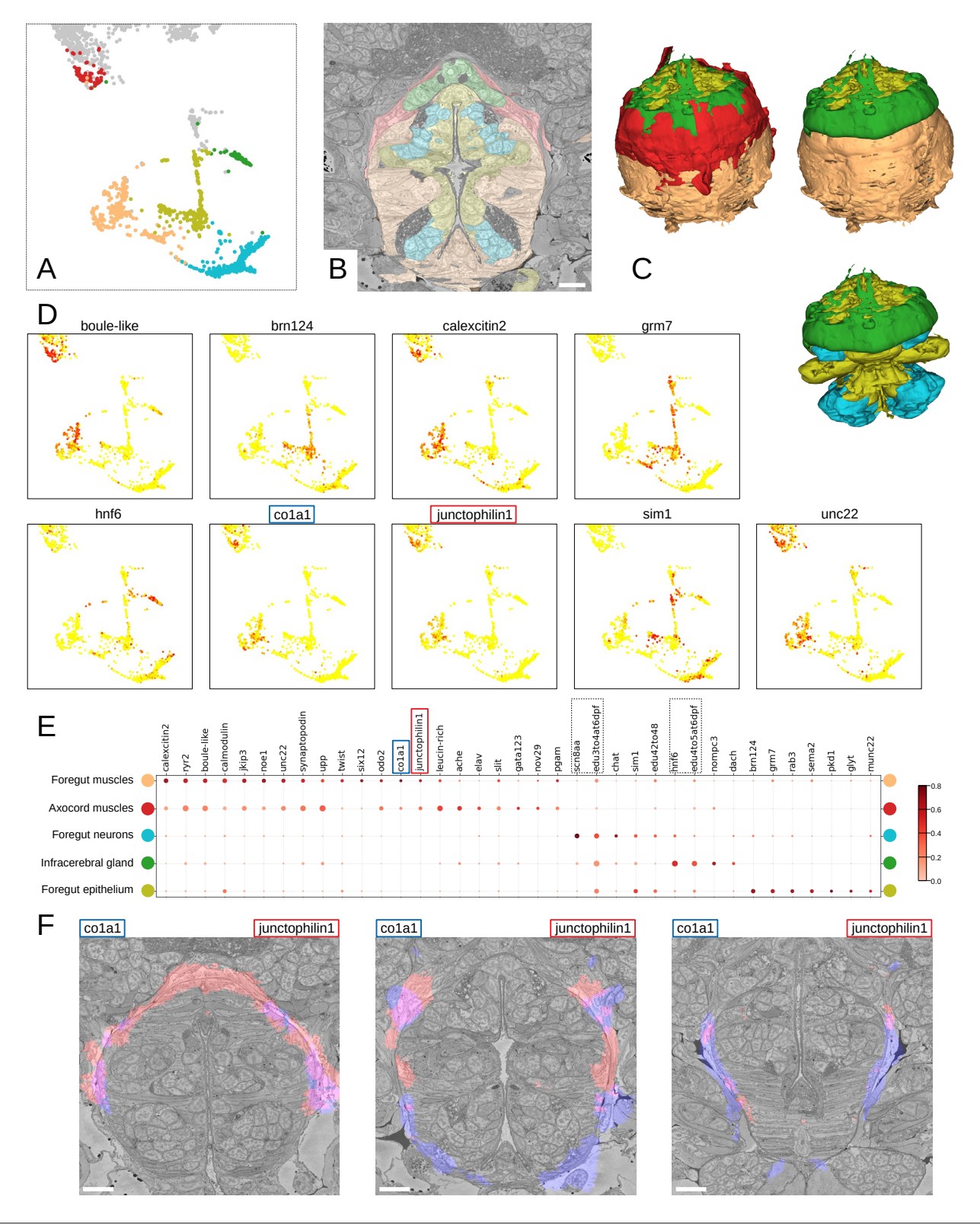

**Figure 9.** Detailed characterisation of the foregut. (**A**) Foregut region clusters plotted on the UMAP representation. (**B**) The foregut tissues, as defined by the MorphoContextFeatures clustering, shown in the animal volume. Scale bar: 10µm. (**C**) A 3D visualisation of the foregut tissues, as defined by the MorphoContextFeatures clustering: all tissues (upper left panel), surrounding muscles removed (upper right panel), both muscle groups removed (lower panel). (**D**) Some of the genes shown to be differentially expressed in the foregut region clusters, plotted on the UMAP representation. (**E**) Gene

*Figure 9 continued on next page*

*Figure 9 continued*

expression dot plot of the foregut region clustering. The size of the dots shows how much of a cluster expresses a gene; the colour shows how much of the expression of a gene is confined to a cluster (see Materials and methods). The genes mentioned in the text are enboxed. (**F**) The genes specific to the axocord muscles, visualised on the animal volume. Scale bars: 10μm.

The online version of this article includes the following figure supplement(s) for figure 9:

**Figure supplement 1.** Foregut clusters on the MorphoFeatures representation.

**Figure supplement 2.** Infracerebral gland.

will be essential to avoid non-biological systematic change in the visual appearance of cells such as intensity shifts, both within and between the EM volumes.

The presented pipeline can also be adjusted for other types of data. For example, since our analysis revealed a high variability in neuronal soma morphologies, it appears reasonable to also apply the pipeline to brain EM volumes, where MorphoFeatures could supplement neuron skeleton features for more precise neuronal morphology description. We also expect the approach to contribute to the analysis of cells in other volumetric modalities, such as X-ray holographic nano-tomography (*Kuan et al., 2020*). For light microscopy with membrane staining, we expect the shape part of the Morpho-Features pipeline to be directly applicable and a useful addition to existing analysis tools. Generally, the versatility of the pipeline is enhanced by its modularity that makes it possible to adapt different parts (e.g. cell shape or nucleus fine texture) to datasets where not all the morphological components are desired or can be extracted.

## Identification of cell types and cell type families with MorphoFeatures

Any attempt to define cell types morphologically has so far been hampered by the selection of relevant properties. What are the morphological features that are best suited for recognising cell types? Regarding neurons, the shapes of dendrites and axons, soma size, and spine density have been measured, in combination with physiological properties including resting potential, biophysical properties, and firing rate (*Zeng and Sanes, 2017*). Volume EM has recently allowed considerable expansion of morphometric parameters, including various shape, intensity, and texture features (*Vergara et al., 2021*). Using these numerous parameters, it is possible to discern general categories such as neurons, muscle, or midgut cells; yet they do not provide further resolution within these groups. Comparing the MorphoFeatures representations to the predefined features (*Vergara et al., 2021*) hints at supreme resolution power that appears to distinguish morphological groupings down to the cell type level (*Figure 4—figure supplement 2*). For example, the subclusters of rhabdomeric photoreceptors of the adult eye and enteric neurons can not be identified in the explicit features representation. Furthermore, the cluster of foregut muscles revealed by the MorphoFeatures appears to be split, with some cells being found closer to striated muscles and some residing between the groups of neurons and dark neurosecretory cells.

Most importantly, the cell clusters identified in MorphoFeature space appear to come closest to genetically defined cell types, as demonstrated here, among others, for different types of mechanosensory cells or cell types of the apical nervous system. Interestingly, the MorphoFeatures also unravel morphological cell types without a genetic match in the cell type atlas (such as the enteric neurons, see above). This is most likely due to the yet limited coverage of the atlas and expected to improve with the continuous addition of genes to the atlas. In particular, the integration of other multimodal omics datasets such as single cell transcriptomics data will be highly rewarding to improve the match between morphological and molecularly defined cell types, which is the ultimate aim of cell type characterisation and categorisation (*Zeng and Sanes, 2017*).

## Towards an unbiased whole-body identification of tissues and organs

The current revolution in the generation of giant volume EM datasets including representations of entire bodies calls for new strategies in the automated processing and comparison of such datasets. This not only involves segmentation and characterisation at the cellular level but also requires the automated identification of discernable structures at the level of tissues and organs. So far, the recognition of tissues in larger volumes was achieved by manual segmentation. We now present a powerful new algorithm to recognise tissues and organs via their co-clustering in the MorphoContextFeature

space, which takes into account similarities between cell neighbourhoods. Using this algorithm, we reproduce and partially outperform tissue recognition via manual segmentation - as exemplified for the ganglionic nuclei of the nereid brain. Beyond that, we present a comprehensive account of foregut tissues, without prior manual segmentation. This analysis reveals a rich collection of tissues including an anterior extension of the axochord surrounding the foregut (*Lauri et al., 2014*) and the infracerebral gland (*Baskin, 1974*; *Golding, 1970*), an adenohypophysis-like neurosecretory tissue that we identify as anterior bilateral extensions of the foregut roof.

While the manual segmentation of tissues mostly relies on the detection of tissue boundaries, our automated tissue detection equally processes the morphological similarities of its constituting cells. This enables tissue separation even in cases where morphological boundaries appear to be absent, as is the case for the anterior and posterior dorsal ganglionic nuclei, which are lacking a clear boundary but are composed of morphologically different cell types that are also genetically distinct (*Vergara et al., 2021*).

## Conclusion

The revolution in multi-omics techniques and volume EM has unleashed enormous potential in generating multimodal atlases for tissues as well as entire organs and animals. The main unit of reference in these atlases is the cell type, which has been defined, and is manifest, both molecularly and morphologically. Our work now provides a versatile new tool to recognise cell types in an automated, unbiased, and comprehensive manner solely based on ultrastructure and shape. For the first time, the MorphoFeatures will allow us to compare and identify corresponding cell types in datasets of the same species, but also between species, on purely morphological grounds.

Furthermore, the MorphoContextFeatures automatically recognise tissues and organs across entire bodies, opening up new perspectives in comparative anatomy. Without previous knowledge, we can obtain the full complement of tissues and organs for a given specimen and quantitatively compare them to those of other specimens of the same or other species. In combination with cell type-, tissue-, and organ-specific expression data, this will enable us to identify corresponding traits in an unbiased manner and with unprecedented accuracy.

## Materials and methods
### Data and preprocessing
### Data

We used the dataset of the marine annelid *P. dumerilii* at late nectochaete stage (6 dpf) from *Vergara et al., 2021*. The dataset comprised an SBEM volume of the complete worm in 10 × 10 × 25 nm resolution, 3D cell and nucleus segmentation masks as well as Whole Mount In-Situ Hybridisation (WMISH) gene expression maps. The segmentation contains 11,402 cells with nuclei that belong to multiple genetically and morphologically distinct cell types. It is important to note that for neurons only somata were segmented since the imaging resolution did not allow for neurite tracing. The gene expression maps cover mainly differentiation genes and transcription factors, which were registered onto the EM volume, exploiting high stereotypy of the *Platynereis* at this stage of development. Gene expression values were also assigned to the EM segmented cells. The maps contain 201 genes and 4 EdU proliferation stainings. We refer to *Vergara et al., 2021* for further details. Since the intensity correction algorithm that was used in the EM volume performed unreliably in regions close to the boundaries of the volume in the z-axis, we excluded all cells located within the region of uncertain intensity assignment quality. These regions correspond to parts of the animal's head and the pygidium.

### Preprocessing

The EM dataset is used to extract the three morphological components separately for cytoplasm and nuclei as follows. Due to the limitations of the GPU memory, for the coarse texture, a central crop is taken from a downsampled version of the data (80 × 80 × 100 nm). The centre of mass of the nucleus is determined, and a bounding box of size 144 × 144 ×144 pixels (cytoplasm) or 104 × 104 × 104 (nuclei) is taken around it. During prediction, the bounding box is increased to 320 × 320 × 320 (cytoplasm) and 160 × 160 × 160 (nuclei) to fit more information. Fine texture patches have a higher resolution (20 × 20 × 25 nm) and are taken separately from cytoplasm and nucleus. For this,

the bounding box surrounding cytoplasm or nucleus is split into cubes of 32 × 32 × 32 pixels, and only those cubes are considered which are less than 50% empty. For both coarse and fine texture, the data is normalised to the range of [0, 1], and for cytoplasm crops, the nuclei are masked out and vice versa.

In order to obtain point clouds and normals of the membrane surfaces, we first converted the segmented 3D voxel volumes into triangular meshes. More specifically, we first applied the marching cubes algorithm *Lorensen and Cline, 1987*; *van der Walt et al., 2014* followed by Laplacian smoothing (*Schroeder et al., 1998*) to obtain smooth triangular surface meshes. These meshes were then simplified and made watertight using *Stutz and Geiger, 2020*; *Cignoni et al., 2008*. Having obtained smooth triangular meshes enables us to compute consistent normals and to apply complex deformations to the underlying surface. During training and inference, the meshes and their surface normals are sampled uniformly to be then processed by the neural network.

## Unsupervised deep-learning pipeline

This section describes our proposed algorithm in more detail. The training and prediction pipeline is implemented in Pytorch (*Paszke et al., 2019*). On a high level, our model consists of six encoder networks which leverage graph convolutional as well as regular convolutional layers in order to learn shape and texture aspects of a cell's morphology. We are using self-supervised learning in combination with stochastic gradient descent in order to obtain good network parameters. The objective that is optimised is a weighted sum of the NT-Xent loss (*Chen et al., 2020*) and the mean squared error and can be interpreted as an unbiased proxy task. We will detail and elaborate on these aspects in the following sections.

### Neural network model

The proposed network, as depicted in (*Figure 1A*), contains six encoders which are encoding shape, coarse, and fine-grained texture of cells and nuclei.

The shape encoder takes as input a set of points in the form of 3D Euclidean coordinates as well as the corresponding normals (see Data and preprocessing) and outputs an N-dimensional shape feature vector. The network itself is a DeepGCN model (*Li et al., 2019*), which is a deep neural network for processing unstructured sets such as point clouds. Conceptually, it consists of three building blocks. In the first block, k-nearest neighbour graphs are built dynamically in order to apply graph convolutions. This can be interpreted as pointwise passing and local aggregation of geometric information which is repeated multiple times. In the second block, the pointwise aggregated geometric information is fused by passing it through a convolutional layer followed by a max pooling operation that condenses the information into a high-dimensional latent vector. Finally, the obtained latent vector is passed through a multilayer perceptron to obtain the shape component of the overall feature vector.

The encoding of fine texture details and coarser contextual information is done using the encoder of a U-Net (*Ronneberger et al., 2015*). The fully convolutional network reduces spatial information gradually by employing three blocks that consist of 3D convolutional layers with ELU activations (*Clevert et al., 2015*) separated by 3DMaxPooling layers. The number of feature maps starts with 64, increases twofold in each block, and afterwards gets reduced to 80 by an additional convolutional layer followed by an ELU activation. In the end, a global average pooling operation is applied to reduce feature dimensionality from 3D to 2D. To ensure that empty regions of the data are not affecting the final feature vector, they are excluded for this final pooling.

### Training

In order to find suitable representations, we leverage self-supervised contrastive learning (CL; *Chen et al., 2020*; *Hadsell et al., 2006*). The idea is to create a representation space in which morphologically similar cells are embedded close to each other, while dissimilar ones are embedded far apart. For learning representations of shape, we follow a purely contrastive approach, whereas for texture representations, we combine the CL objective with an autoencoder reconstruction loss. Both are described in more detail below.

The training procedure for CL is the same for shape and texture representations. First, we randomly sample a mini batch of cells or nuclei. Two augmented views are then created for each sample in the batch (see Data augmentations) and concatenated to form an augmented batch of twice the size. The augmented batch is passed through the neural network encoder to compute representations. Those

representations that share the same underlying sample are considered as positive pairs, whereas the others are considered as negative pairs. The NT-Xent loss (*Chen et al., 2020*) is then used to score the representations based on their similarity and, therefore, encourages the network to embed positive pairs closer to each other than negative pairs.

For learning shape representations, we first augmented all cell and nucleus meshes using biharmonic deformation fields and as-rigid-as-possible transformations in a preprocessing step (see Data augmentation). More specifically, we computed 10 random views for each membrane surface. The augmented meshes are then used for training, i.e., we compute surface normals and sample 1024 points and their corresponding normals which form the inputs of the shape encoders. In each training iteration, we sampled 48 cells/nuclei, which resulted in a batch size of 96. Adam (*Kingma and Ba, 2014*) with a learning rate of 0.0002 and a weight decay of 0.0004 was used to update the network parameters. The hyperparameters were obtained by a random search.

In order to learn representations of texture, we combine the NT-Xent loss with an autoencoder reconstruction objective. The latent representations, taken before the dimensionality reduction step of global average pooling, are further processed by a symmetric decoder part of the UNet network with only one block, which aims to reconstruct the network inputs. The reconstructions are then compared to the original inputs and scored by a mean squared error loss. Additionally, an L2-Norm loss was applied to flattened bottleneck features to restrict the range of possible values. The final loss is thus a weighted sum of the NT-Xent, the mean squared error loss, and the L2-Norm loss. The batch size was set to 12 and 16 for cell and nuclei coarse texture and to 32 for fine texture. For training, we used Adam (*Kingma and Ba, 2014*) as optimizer with a learning rate of 0.0001 and a weight decay of 0.00005. The model was evaluated every 100 iterations, and the best result was saved with a patience of 0.95. The learning rate was reduced by 0.98 whenever the validation score did not show improvement with a patience of 0.95.

## Data augmentation

A key component for successful contrastive representation learning are suitable data augmentations. In our context, applying augmentations to a cell can be interpreted as creating morphological variations of a cell. By creating positive and negative pairs (cf. previous section), the network learns to distinguish true variations from noise and non-essential ones. It is therefore important to find a suitable set of transformations that distort the right features of a cell and do not change its morphological identity.

As augmentations for shape, we leveraged biharmonic and as-rigid-as-possible transformations (*Botsch and Kobbelt, 2004*; *Sorkine and Alexa, 2007*, *Jacobson et al., 2018*) as they deform cellular and nuclear membranes in a way that retains most characteristic geometric features of the original surface. In order to compute these deformations, we sample so-called handle regions on the membrane surface. These regions are translated in a random, normal, or negative normal direction. The deformation constraint is then propagated to the rest of the surface either via a biharmonic deformation field or an as-rigid-as-possible deformation. We combine these deformations with random anisotropic scaling, rotations, and/or symmetry transformations.

To create positive samples for the contrastive training of texture, we used random flips and rotations and elastic transformations of the data.

## Inference

In order to compute the MorphoFeatures representation for the whole animal, we processed each cell's cellular and nuclear shape and texture, as described in previous sections, with our trained networks. For the fine texture patches to get one feature vector per cell, the feature vectors of all the patches from a cell were averaged. Each cell is represented by a 480-dimensional vector that consists of a nuclear and cellular part. Each of these parts consists of a shape, coarse, and fine-grained texture 80-dimensional feature vector. This vector size was found empirically using the bilateral distance as a metric. Vectors of bigger size showed similar performance, but contained more redundant features.

## Quantitative feature evaluation

In this section, we describe our feature evaluation experiments in more detail. In order to assess the quality of MorphoFeatures quantitatively, we conducted cell-type prediction and bilateral pair analysis

experiments. We use the same set of explicit features that were used in *Vergara et al., 2021* for analysing cellular morphology as a baseline for comparisons.

## Explicit feature baseline

The explicitly defined representation includes shape features (volume in microns, extent, equivariant diameter, major and minor axes, surface area, sphericity, and max radius), intensity features (different quantiles of intensity mean, median, and SD), and Haralick texture features. These features were calculated, whenever appropriate, for cell cytoplasm, nucleus, and chromatin segmentation. The calculated features were taken from *Vergara et al., 2021*: https://github.com/mobie/platybrowser-project/blob/main/data/1.0.1/tables/sbem-6dpf-1-whole-segmented-cells/morphology.tsv for cellular segmentation and from https://github.com/mobie/platybrowser-project/blob/main/data/1.0.1/tables/sbem-6dpf-1-whole-segmented-nuclei/morphology.tsv for nucleus and chromatin segmentation. The UMAP of these features was taken from https://github.com/mobie/platybrowser-project/blob/main/data/1.0.1/tables/sbem-6dpf-1-whole-segmented-cells/morphology_umap.tsv.

## Classification of visibly distinct groups of cells

In order to verify that MorphoFeatures can distinguish visibly different groups of cells, we evaluated the prediction accuracy of a logistic regression model that was trained on a small dataset of 390 annotated cells taken from *Vergara et al., 2021* (*Figure 2A*). The original annotation contained 571 neurons, 141 epithelial cells, 55 midgut cells, 53 muscle cells, 51 secretory cells, 41 dark cells, and 25 ciliated cells. We first corrected the assignments deemed incorrect by the annotators with expertise in *Platynereis* cell types. Then for the classes that contained less than 60 cells, we manually annotated more cells. From bigger classes, we selected 60 cells, selecting the most morphologically diverse ones when possible. For ciliated cells, we were only able to locate 30 cells in the whole animal.

First, we computed the MorphoFeatures representation for each cell in the animal (excluding cells with severe segmentation errors) and standardised features across the whole dataset. We then trained and validated a logistic regression model using a stratified k-folds cross validator with fivefolds. We used the implementations of logistic regression, stratified k-fold from Scikit-learn (*Pedregosa et al., 2011*) with the following arguments: C=1, solver = 'lbfgs'. To prevent potential overfitting due to the high number of features, we used feature agglomeration (*Pedregosa et al., 2011*; n_clusters = 100) to reduce the number of similar features.

## Bilateral pair analysis

The criterion of classifying distinct groups of cells in the previous section can be seen as a coarse and global way of evaluating MorphoFeatures. In this section, we describe how we used the animal's bilateral symmetry to have a more fine-grained measure for quantitatively evaluating nuanced morphological differences. The measure, which we will refer to as bilateral distance, was introduced in *Vergara et al., 2021* and leverages the fact that most cells in the animal have a symmetric partner cell on the other side of the mediolateral axis. The bilateral distance is computed in the following way: first, we determine for each cell $c$ a group of N potential partner cells $\{c_1, , c_N\}$ on the opposite side. We then compute a ranking based on the Euclidean distances $\|c - c\|$ in feature space between cell $c$ and all other cells $c \neq c$. Since $c$'s symmetric partner has the same morphology, up to natural variability, it should be among the closest cells in the ranking. We then determine the highest ranked partner cell as $\mathrm{rank}(\mathrm{argmin}_{n \in \{1,\dots,N\}} \|c - c_n\|)$. In doing this for all cells, we create a list containing the ranking positions of cells to their symmetric partner. The bilateral distance is then defined as the median of this list and can be interpreted as a measure of how close two morphologically very similar cells are on average in feature space.

## Adding neighbour morphological information

This section describes how neighbourhood information was incorporated into MorphoFeatures in order to obtain MorphoContextFeatures. First, we built a region adjacency graph from the cell segmentation masks using scikit-image (*van der Walt et al., 2014*). The obtained graph reflects the animal's cellular connectivity, i.e., cells are represented by nodes in the graph, and nodes are connected by an edge if the corresponding cells are neighbours. We then assign node attributes by mapping the MorphoFeatures representation of a cell to the corresponding node. Neighbourhood

features are then obtained by computing the mean over a node's MorphoFeatures vector and the MorphoFeatures vectors of its surrounding neighbours. Finally, a cell's neighbourhood features are concatenated with the cell's MorphoFeatures to obtain the MorphoContextFeatures representation.

We benchmarked multiple methods for aggregating feature vectors of neighbouring cells, including taking the mean, the median, and approaches based on graph neural networks. Among these methods, taking the mean showed to be the best tradeoff between performance and simplicity. Compared to taking the median, the mean features showed to give a higher accuracy on predicting morphotypes and have a lower bilateral distance. The features extracted by graph neural networks exhibited unstable behaviour, both in predictiveness of morphotypes as well as bilateral distance.

## Feature analysis

### UMAP representation analysis

Visualisation was done using the UMAP package (*McInnes et al., 2018*) with the following parameters: Euclidean metric, 15 neighbours, and 0 minimal distance. Cell metadata, such as annotated types, animal regions, or gene expression, was plotted on the resulting UMAP.

### Clustering

Cell clustering was done according to the following procedure. First, the features were standardised to zero mean and unit variance. Then, following *Dubourg-Felonneau et al., 2021*, a weighted k-neighbour graph was constructed from MorphoFeatures representations of all the animal cells using the UMAP algorithm (*McInnes et al., 2018*) with the following arguments: n_neighbours = 20, metric='Euclidean', and min_dist = 0. Afterwards the Leiden algorithm (*Traag et al., 2019*) for community detection was used to partition the resulting graph, using the CPMVertexPartition method and a resolution parameter of 0.004. For more fine-grained clustering, the resolution was increased to 0.1 to split apart secretory and ciliary band cells, 0.2 to split photoreceptors and enteric neurons, 0.07 to split the midgut cluster, 0.005 for muscle neighbourhood cluster (n_neighbours for the UMAP clustering was adjusted to 10), and 0.01 for splitting foregut neurons and epithelial cells.

One of the clusters was discarded since visual examination showed it contained only cells with split segmentation errors that 'cut' through the nucleus area along one of the axes (examples shown in *Figure 3—figure supplement 1*). Other types of segmentation errors in the dataset mostly did not influence the clustering quality for multiple reasons. First, most errors are rather randomly distributed and not consistent, meaning the size of falsely attached fragments is highly variable, and the location of segmentation splits is not consistent. Secondly, many of the errors are cell type specific. For example, midgut cells often got merged with midgut lumen, (*Figure 3—figure supplement 2*, dark green) and muscle cells often experienced splits or merges of other muscle pieces (*Figure 3—figure supplement 2*, dark and light orange). Small merges that are not cell type specific (*Figure 3—figure supplement 2*, light green) also minimally affected the assignment. However, substantial merges could lead to a cell being grouped with a different cell type, for example, a neuron that got merged with a piece of epithelial cell bigger than the neuron size (*Figure 3—figure supplement 2*, yellow-green) got grouped together with epithelial cells, not neurons.

To visualise average shape and texture of a cluster (4B,C), a median feature vector was computed across the cluster, and the cell with the feature vector closest to the median one was taken as the average cell.

### Gene expression analysis

To produce the cluster gene expression 'dot plots' (*Figures 4A, 5B, 8C and 9E*), the gene expression values for each cell were taken from *Vergara et al., 2021* at https://github.com/mobie/platy-browser-project/blob/main/data/1.0.1/tables/sbem-6dpf-1-whole-segmented-cells/genes.tsv. Similar to *Vergara et al., 2021*, for each cluster and for each gene, we calculated three values: (A) the mean expression in the cluster, (B) the fraction of the total animal/region expression within this cluster (the sum of expression in this cluster divided by the sum of the expression in the whole animal or the selected region), and (C) the specificity defined as $C = 2AB/(A + B)$. Intuitively, $A$ indicates how much of the cells in a given cluster express a given gene, $B$ shows how selective this gene for this cluster is, and $C$ is a harmonic mean of the two values. To create a dot plot, only the genes with $C$ value above 0.15 were used. For the neuron clusters 1, 3, and 5 (*Figure 4A*), the threshold was increased to 0.2 due

to a high number of specifically expressed genes available. For the midgut clusters (*Figure 5*) to show only the genes differing between the two clusters, the genes were removed, where the specificity was above 0.15 in both clusters but differed by less than 0.4. The size of the dots in the plots reflects $A$, and the colour corresponds to $B$. The cluster with the most gene expression was determined, and the remaining clusters were sorted by their similarity to it. To illustrate some of the genes showing differential expression in these groups (*Figures 5, 8 and 9*), the corresponding regions of the UMAP representation were cut out, and these genes were plotted on top.

## Feature specificity analysis

The feature specificity 'dot plots' were generated as follows. The MorphoFeatures of all cells were processed by first clipping the values below –2 and above +2 (less than 5% of values affected) and standardised to the range of [0, 1]. Then three values $A$, $B$, and $C$ were calculated similarly to gene expression dot plots. However, since smaller clusters would have smaller sum of any feature, we additionally normalised the value $B$ by diving it by the relative cluster size. To create a dot plot, only the features with $C$ value above 1 were used.

To illustrate some features specific to clusters (*Figure 6—figure supplement 1*), we first selected a set of features from the dot plot with high specificity in a given cluster and low specificity in other clusters. We then showed their extremes with cells that have the respective minimal/maximal value of this feature (*Figure 6*). The cells with merge errors of more than half of a cell size were ignored. In order to visualise the 3D cells and nuclei, we selected a 2D view that we presume shows the shape or texture feature common between all four selected cells with minimal or maximal value of the feature.

Visualising the extremes of various features from the dot plot, we found most features (80–90%) to be relatively easy to interpret. More specifically, 82% shape features, 80% fine texture features, and 90% coarse texture features could be visually mapped to explainable properties in less than 2 min of observing the data. However, when selecting a random subset (10–20 random features) from each morphological component, the number of clearly interpretable features was lower, 65% for fine, 70% for coarse texture, and 45% for shape features.

## Ganglionic nuclei

To compare ganglionic nuclei (*Figure 8*), we used manually segmented regions from https://github.com/mobie/platybrowser-project/blob/main/data/1.0.1/tables/sbem-6dpf-1-whole-segmented-cells/ganglia_ids.tsv and https://github.com/mobie/platybrowser-project/blob/main/data/1.0.1/tables/sbem-6dpf-1-whole-segmented-cells/regions.tsv and gene clusters from https://github.com/mobie/platybrowser-project/blob/main/data/1.0.1/tables/sbem-6dpf-1-whole-segmented-cells/gene_clusters.tsv (*Vergara et al., 2021*).

## Data visualisation

We used matplotlib (*Hunter, 2007*) and seaborn (*Waskom, 2021*) for plotting the data, MoBIE (*Pape et al., 2022*) for visualising cells in the EM volume, and vedo (*Musy et al., 2022*) for visualising meshed cells and tissues.

## Acknowledgements

We thank all the members of the Kreshuk and Uhlmann group, especially Constantin Pape, for helpful comments. We further thank Christian Tischer for implementing features essential for biological analysis and exploration in MoBIE. We thank Hernando M Vergara for assistance in cell type recognition and Kimberly I Meechan for help with comparison to explicit features. We would also like to acknowledge the support of the EMBL IT Services, especially Juri Pecar for maintaining the GPU resources. The work was supported by EMBL internal funds and by a grant from the European Research Council (NeuralCellTypeEvo 788921) to DA.

## Additional information

### Funding

| Funder | Grant reference number | Author |
| --- | --- | --- |
| European Research Council | 788921 | Detlev Arendt |

The funders had no role in study design, data collection and interpretation, or the decision to submit the work for publication.

### Author contributions

Valentyna Zinchenko, Conceptualization, Software, Formal analysis, Validation, Investigation, Visualization, Methodology, Writing - original draft, Writing - review and editing; Johannes Hugger, Software, Formal analysis, Validation, Visualization, Methodology, Writing - original draft, Writing - review and editing; Virginie Uhlmann, Conceptualization, Supervision, Methodology, Writing - review and editing; Detlev Arendt, Conceptualization, Supervision, Methodology, Writing - original draft, Writing - review and editing; Anna Kreshuk, Conceptualization, Supervision, Funding acquisition, Methodology, Writing - original draft, Writing - review and editing

### Author ORCIDs

Valentyna Zinchenko (ID) http://orcid.org/0000-0001-6900-0656
Detlev Arendt (ID) http://orcid.org/0000-0001-7833-050X
Anna Kreshuk (ID) http://orcid.org/0000-0003-1334-6388

### Decision letter and Author response

Decision letter https://doi.org/10.7554/eLife.80918.sa1
Author response https://doi.org/10.7554/eLife.80918.sa2

## Additional files

### Supplementary files

- MDAR checklist

### Data availability

The study relies on the publicly available data from *Vergara et al., 2021*. The generated cell representations are available on GitHub: https://github.com/kreshuklab/MorphoFeatures.

The following previously published dataset was used:

| Author(s) | Year | Dataset title | Dataset URL | Database and Identifier |
| --- | --- | --- | --- | --- |
| Vergara P, Meechan Z | 2021 | Whole-body integration of gene expression and single-cell morphology | https://www.ebi.ac.uk/empiar/EMPIAR-10365/ | EMPIAR, 10365 |

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
