## [Editor Report]

This paper marks a fundamental advance in reconstruction of volume EM images, by introducing the automatic assignment of cell types and tissues. This task has previously been done manually, resulting in a serious bottleneck in reconstruction, but the authors present compelling evidence that in at least some cases, automatic and semi-automatic techniques can match or better human assignment of cell and tissue types. These results will be of interest to almost all groups doing EM reconstruction, as they can speed up cell type assignment when the cell types are known, and provide an initial cell type and tissue classification when they are not.

---

## [Decision Letter]

**Decision letter after peer review:**

Thank you for submitting your article "MorphoFeatures: unsupervised exploration of cell types, tissues and organs in volume electron microscopy" for consideration by *eLife*. Your article has been reviewed by 2 peer reviewers, including Louis K Scheffer as Reviewing Editor and Reviewer #1, and the evaluation has been overseen by Claude Desplan as the Senior Editor. The following individual involved in review of your submission has agreed to reveal their identity: Jan Funke (Reviewer #2).

Essential revisions:

1) What happens when there are errors in segmentation? This can easily lead to shapes which are different from any legitimate cell type, and could result in an erroneous cell type being created. On the other hand, these shapes are not likely reproduced, and so would likely result in a cluster of a single cell, which would be suspicious (at least in bilaterally symmetric creatures) as every cell type should have at least two examples.

It would be good to show the results of an existing merge error (or induce one if necessary) and report the results.

2) Along similar lines, the paper should report the used (or potential) cell-typing flow when using this method. In the paper it speaks of manually correcting the segmentation. But how do you know which cells are wrong? If you need to examine each cell manually, there will not be much time savings. So possibly you segment, then run the algorithms of this paper. Then you look for clusters of size 1, which (assuming bilateral symmetry) are likely a mistake. Then you fix those cells and iterate. It would be great to know if this approach was used, and if so how fast it converges.

3) In the section on "visually interpretable" features, you should provide a more quantitative idea of how many features are considered meaningful, and how those can be found. For example, are the six features shown in Figure 3 particularly meaningful, or were they chosen among many? A discussion of the feature selection protocol would be useful for replicating the method on new data. Furthermore, a supplementary figure with some of the features which are not meaningful would give the reader a better idea of the range of interpretability to expect.

4) The section on MorphoContextFeatures is missing a comparison with the MorphoFeatures. This made it unclear to me whether adding the neighborhood information is necessary for the discovery of tissues and organs. This could be remedied with a supplementary figure showing the same analysis as in figures 7 and 8 on the MorphoFeatures without the additional neighborhood information. Alternatively, since the MorphoFeatures are a subset of the MorphoContextFeatures, the authors could run a post-hoc analysis of whether the MorphoFeatures or the neighborhood features best explain the inter-class variance.

5) Finally, some extra guidance is needed to replicate this work on new data. In particular the following points could use more discussion:

5.1. How to choose the size of the MorphoFeatures vector – did the authors attempt a number other than 80 and if so, what was affected by this choice?

5.2. The protocol for when and how to define sub-clusters – were the chosen thresholds based on prior knowledge such as known tissues/organs? What do the authors suggest if this kind of information is missing?

5.3 How to link the obtained clusters back to specific, potentially meaningful, MorphoFeatures. For example, does the distinctive shape of the enteric neurons in cluster 8.3 of figure 5 correspond to an extreme of the cytoplasm shape feature described in figure 3 (lower left)?

6) In figure 1 b/c: The difference between B and lower part of C is unclear. If seen as a description of the two losses, the fact that the contrastive loss is shown twice is confusing. If seen as a description of the whole training setup, the omission of the fine-grained features is the issue.

7) In figure 2: It would be interesting to find out which subset of features correlates with which class, and whether those are meaningful. At minimum, knowing whether a shape, coarse texture, and fine texture are all involved in predictions.

8) In figure 2: The legend on the scale bars says 'mkm', which is not an abbreviation of micrometers that I am used to. Perhaps μm instead? The legend is also difficult to see (see pt. 11).

9) In figure 5: the scale bar legend is too small to see. Also, putting it above the scale bar might improve readability.

10) In Figure 7 + text: the text suggests that the clusters have been chosen manually, rather than using community detection as in the other figures. This should be justified if true.

11) In figure 8B + text (p.14): There isn't much said about the cells that are misclassified in the MorphoContextFeatures, i.e. where both manual segmentation and gene clusters agree, but MorphoContextFeatures does not. For example: green cell among the brown, or yellow cells just right of the central line of the organism, top. A justification similar to the explanations of misclassification in Figure 2 would help strengthen the argument.

For future work: Currently many EM reconstructions are nervous systems of somewhat higher animals (*Drosophila* and small portions of mammal brains). The shapes of these cells are very complex, and it would be interesting to see if the morphology features will continue to work on such complex cells. *Drosophila* could be a good example.

There is a question (line 409) of how well patch characteristics will correspond when comparing different samples. This could be tested, at least in part, by applying different image normalizations to the same sample, then treating them as two separate samples.

*Reviewer #1 (Recommendations for the authors):*

I have two main technical concerns. The first is what happens when there are errors in segmentation. This can easily lead to shapes which are different from any legitimate cell type, and could result in an erroneous cell type being created. On the other hand, these shapes are not likely reproduced, and so would likely result in a cluster of a single cell, which would be suspicious (at least in bilaterally symmetric creatures) as every cell type should have at least two examples. It would be good to induce a (for example) merge error between two cells and see what happens.

I would be very interested in the cell-typing flow using this method. In the paper it speaks of manually correcting the segmentation. But how do you know which cells are wrong? If you need to examine each cell manually, there will not be much time savings. So I could imagine a flow where you segment, then run the algorithms of this paper. Then you look for clusters of size 1, which (assuming bilateral symmetry) are likely a mistake. Then you fix those cells and iterate. It would be great to know if this approach was used, and if so how fast it converges.

For future work: Currently many EM reconstructions are nervous systems of somewhat higher animals (*Drosophila* and small portions of mammal brains). The shapes of these cells are very complex, and it would be interesting to see if the morphology features will continue to work on such complex cells. *Drosophila* could be a good example.

There is a question (line 409) of how well patch characteristics will correspond when comparing different samples. This could be tested, at least in part, by applying different image normalizations to the same sample, then treating them as two separate samples.

*Reviewer #2 (Recommendations for the authors):*

1. In figure 1 b/c: The difference between B and lower part of C is unclear. If seen as a description of the two losses, the fact that the contrastive loss is shown twice is confusing. If seen as a description of the whole training setup, the omission of the fine-grained features is the issue.

2. In figure 2: It would be interesting to find out which subset of features correlates with which class, and whether those are meaningful. At minimum, knowing whether a shape, coarse texture, and fine texture are all involved in predictions.

3. In figure 2: The legend on the scale bars says 'mkm', which is not an abbreviation of micrometers that I am used to. Perhaps μm instead? The legend is also difficult to see (see pt. 4).

4. In figure 5: the scale bar legend is too small to see. Also, putting it above the scale bar might improve readability.

5. In Figure 7 + text: the text suggests that the clusters have been chosen manually, rather than using community detection as in the other figures. This should be justified if true.

6. In figure 8B + text (p.14): There isn't much said about the cells that are misclassified in the MorphoContextFeatures, i.e. where both manual segmentation and gene clusters agree, but MorphoContextFeatures does not. For example: green cell among the brown, or yellow cells just right of the central line of the organism, top. A justification similar to the explanations of misclassification in Figure 2 would help strengthen the argument.

---

## [Author Response]

Essential revisions:1) What happens when there are errors in segmentation? This can easily lead to shapes which are different from any legitimate cell type, and could result in an erroneous cell type being created. On the other hand, these shapes are not likely reproduced, and so would likely result in a cluster of a single cell, which would be suspicious (at least in bilaterally symmetric creatures) as every cell type should have at least two examples.It would be good to show the results of an existing merge error (or induce one if necessary) and report the results.

This is an interesting question, however, there is no straightforward answer, as it depends on the type of the error and its abundance. In our data there is only one example of a segmentation error forming a separate cluster. These are 26 cells with a prominent shape phenotype, ‘cut’ through the nucleus area along one of the axes (examples now shown in Figure 3—figure supplement 1). Such cuts triggered extremely high values of some cell shape features, affecting the MorphoFeatures of these cells severely enough to result in a new cluster.

However, other types of segmentation errors do not form separate clusters for multiple reasons. Firstly, most errors are rather randomly distributed and not consistent, meaning the size of falsely attached fragments is highly variable and the location of segmentation splits is not consistent. For the cells of one class this often creates a ‘continuum’ of possible segmentation errors, which will get clustered together. Secondly, many errors are actually cell type specific. This comes from the fact that the segmentation training data contained mostly neurons, thus, other cell types further away from the training data suffered more segmentation errors. For example, muscles often experience split errors or get merged with a piece of neighbouring muscle and midgut cells often get merged with midgut lumen (Figure 3—figure supplement 2). Given random distribution of such splits and merges, they do not form separate clusters, but rather get clustered with the cells of the same cell type.

Finally, the modularity of the pipeline makes it to some extent robust to many segmentation errors, including relatively big merges (as shown in Figure 3—figure supplement 2). Such errors rarely affect nuclei segmentation, which is reported to have 99.0% accuracy (Vergara, 2021). That means that even severe cell segmentation errors will not affect at least 50% of MorphoFeature vector. In practice this number is often higher, since cell texture features will be minimally affected by split errors, and merge errors will only affect them proportionally to the size of the merged part. The only part of the feature vector heavily affected by segmentation errors is cell shape. However, since it comprises only _⅙_ of the full feature vector, unique shapes will not necessary result in feature vectors distinct enough to create a separate cluster.

It is also important to note that the original EM dataset has been segmented using relatively standard segmentation approaches ((Beier, 2017) with modifications from (Pape, 2019)). Thus, although we can not comment on how the pipeline will behave for other types of segmentation errors, we expect similar errors to occur in similarly segmented datasets.

In the revised version of the manuscript, we now comment on the influence of segmentation errors in the Methods section (line 719):

“One of the clusters was discarded, since it contained only cells with split segmentation errors that ‘cut’ through the nucleus area along one of the axes (examples shown in Figure 3—figure supplement 1). Other types of segmentation errors in the dataset mostly did not influence the clustering quality for multiple reasons. Firstly, most errors are rather randomly distributed and not consistent, meaning the size of falsely attached fragments is highly variable and the location of segmentation splits is not consistent. Secondly, many of the errors are cell type specific. For example, midgut cells often got merged with midgut lumen (Figure 3—figure supplement 2, dark green) and muscle cells often experienced splits or merges of other muscle pieces (Figure 3—figure supplement 2, dark and light orange). Small merges that are not cell type specific (Figure 3—figure supplement 2, light green) also minimally affected the assignment. However, substantial merges could lead to a cell being grouped with a different cell type, for example, a neuron that got merged with a piece of epithelial cell bigger than the neuron size (Figure 3—figure supplement 2, yellow-green) got grouped together with epithelial cells, not neurons.”

2) Along similar lines, the paper should report the used (or potential) cell-typing flow when using this method. In the paper it speaks of manually correcting the segmentation. But how do you know which cells are wrong? If you need to examine each cell manually, there will not be much time savings. So possibly you segment, then run the algorithms of this paper. Then you look for clusters of size 1, which (assuming bilateral symmetry) are likely a mistake. Then you fix those cells and iterate. It would be great to know if this approach was used, and if so how fast it converges.

Based on multiple reviewers comments, we have introduced a separate guide to the potential application of our method to new data (see revision point 6). As a part of it, we also describe the cell-typing flow. However, we do not comment on manual segmentation correction, since it was done in (Vergara, 2021) and was not part of our paper. As discussed above, our pipeline is relatively robust to small segmentation errors, however, we generally expect that substantial merge errors (more than ~25% of a cell attached) are corrected before applying it.

3) In the section on "visually interpretable" features, you should provide a more quantitative idea of how many features are considered meaningful, and how those can be found. For example, are the six features shown in Figure 3 particularly meaningful, or were they chosen among many? A discussion of the feature selection protocol would be useful for replicating the method on new data. Furthermore, a supplementary figure with some of the features which are not meaningful would give the reader a better idea of the range of interpretability to expect.

The reviewers raise a very important point here, however, we find some parts of it are out of the scope of our paper. Specifically, to calculate the percentage of meaningful features, we must first define what ‘meaningful’ means. Some features might be difficult to understand as a human, yet be essential for distinguishing different types of morphologies. That is why in the paper we rather talk about visual interpretability. This characteristic, however, is not binary, with some features being easier to interpret than others. For example, for visualisation purposes we looked for clearly understandable features, but many of the features we did not include in the figure could still be visually explained after a more detailed examination. So while it is easy to define clearly interpretable features, confidently selecting some that are not interpretable at all is barely possible. That is why we now rather report the approximate percentage of ‘easily visually interpretable’ features in the Methods section (line 767):

“Visualising the extremes of various features from the dot plot we found most features (80-90%) to be relatively easy to interpret. More specifically, 82% shape features, 80% fine texture features and 90% coarse texture features could be visually mapped to explainable properties in less than 2 minutes of observing the data.

However, when selecting a random subset (10-20 random features) from each morphological component, the number of clearly interpretable features was lower, 65% for fine and 70% for coarse texture and 45% for shape features.”

Addressing the revision comment 6.3, we also changed the way we select the features to visualise, which we now describe in the Methods section (line 760):

“To illustrate some features specific to clusters (Figure 6—figure supplement 1) we first selected a set of features from the dot plot with high specificity in a given cluster and low specificity in other clusters. We then showed their extremes with cells that have the respective minimal/maximal value of this feature (Figure 6). The cells with merge errors of more than half of a cell size were ignored.”

4) The section on MorphoContextFeatures is missing a comparison with the MorphoFeatures. This made it unclear to me whether adding the neighborhood information is necessary for the discovery of tissues and organs. This could be remedied with a supplementary figure showing the same analysis as in figures 7 and 8 on the MorphoFeatures without the additional neighborhood information. Alternatively, since the MorphoFeatures are a subset of the MorphoContextFeatures, the authors could run a post-hoc analysis of whether the MorphoFeatures or the neighborhood features best explain the inter-class variance.

We thank the reviewers for raising this interesting point. We have further investigated the possibilities created by adding the neighbourhood information and introduced two new Supplementary Figures to show the value of MorphoContextFeatures (line 355):

“To further examine the contribution of incorporating cellular neighbourhood morphology we visualise the discovered foregut tissues and ganglia on the MorphoFeatures representation (Figure 8—figure supplement 1 and Figure 9—figure supplement 1). This shows that while some of the tissues, such as palpal ganglia or foregut epithelium, are composed of cells of one morphological type, others, e.g. circumpalpal ganglia or infracerebral gland, comprise cells with different morphology and could only be detected when taking into account cellular context as well.”

5) Finally, some extra guidance is needed to replicate this work on new data. In particular the following points could use more discussion:

We thank the reviewers for these suggestions that would highly improve the practical value of the paper. We wrote a guide that describes the steps necessary to apply the pipeline to a new dataset. However, given that in order to replicate the work one would first have to refer to the code in our GitHub repository, we have decided that it would be most appropriate to place the guide there:

https://github.com/kreshuklab/MorphoFeatures/blob/main/morphofeatures/howto_new_dataset.md.

We invite the reviewers to kindly advise us on whether they believe this guide should also be placed in the paper text, and if so, where exactly.

5.1. How to choose the size of the MorphoFeatures vector – did the authors attempt a number other than 80 and if so, what was affected by this choice?

We now briefly mention the reasoning behind this feature vector size in the Methods section (line 635):

“This vector size was found empirically using the bilateral distance as a metric. Vectors of bigger size showed similar performance, but contained more redundant features.”

We also discuss how to choose it for new data in the abovementioned guide:

“In case of no proxy task available, which could be used to evaluate different sizes, we recommend rather choosing a bigger vector size (e.g. 512). The feature vector could be then compressed to remove highly correlated features using, for example, feature clustering or dimensionality reduction techniques.”

5.2. The protocol for when and how to define sub-clusters – were the chosen thresholds based on prior knowledge such as known tissues/organs? What do the authors suggest if this kind of information is missing?

We now discuss the protocol for defining clustering resolution in the guide:

Our algorithm for setting this parameter and further defining subclusters was as follows:

– Choose an initial parameter that would result in a smaller number of clusters that could be easily analysed manually (10-20).

– Visually inspect the clusters and further subcluster (by increasing clustering resolution) if necessary. We suggest further subclustering in the following cases:

– cells in a cluster occupy clearly distinct areas in the volume (e.g. subclusters 8.1, 8.2 and 8.3 in Figure 3B);

– cells have distinct visual appearance (e.g. subclusters 14 and 15 in Figure 3B);

– gene expression (or other available information about the data) shows high heterogeneity in a cluster (subclustering midgut cells, Figure 5);

– prior knowledge is available, such as cell types or broader cell classes (e.g. splitting the cluster of foregut neurons and foregut epithelial cells, Figure 9).

5.3 How to link the obtained clusters back to specific, potentially meaningful, MorphoFeatures. For example, does the distinctive shape of the enteric neurons in cluster 8.3 of figure 5 correspond to an extreme of the cytoplasm shape feature described in figure 3 (lower left)?

We are grateful to the reviewers for suggesting this interesting analysis. We investigated the specificity of different features in morphological clusters and added a new Figure 6—figure supplement 1, which shows a dot plot of feature specificity in the clusters. We further changed the section “MorphoFeatures correspond to visually interpretable morphological properties” to discuss linking morphological clusters to specific MorphoFeatures and changed the figure in this section to show visual interpretation of some of the features specific to the clusters of outer epithelial cells, enteric neurons, midgut cells, rhabdomeric photoreceptors, ciliary band cells and foregut muscles (line 297):

“To examine whether it is possible to understand which properties of cells learned by our network distinguish the discovered morphological groups, we first identified MorphoFeatures that have high values specific to each cluster (Figure 6—figure supplement 1). Then for a set of characteristic features we visualised cells that correspond to the maximum and minimum value of the corresponding feature (Figure 6). Visual inspection showed that many of them can be matched to visually comprehensible properties.

For example, cytoplasm coarse texture feature 21 (Figure 6, upper left), which distinguishes the outer epithelial cells (cluster 11), shows its minimal value in secretory cells, which contain multiple highly distinct types of texture, and its maximal value in the cells crowded with small vesicles and cisternae on the surface of the animal. The cluster of enteric neurons (cluster 8.3) strongly differs from other neurons by nuclear coarse texture feature 4 (Figure 6, upper right), which contrasts nuclei with large amount of heterochromatin often attached to the nuclear lamina to nuclei with a high amount of euchromatin and prominent nucleoli. Cytoplasm fine texture feature 50 (Figure 6, middle left), characteristic to the midgut cells (cluster 14) is the lowest in cells with a small stretch of homogeneous cytoplasm with mitochondria and has its peak in cells with abundant Golgi cisternae and medium-sized mitochondria. Rhabdomeric photoreceptors of the adult eye (cluster 8.1) display specific nuclear fine texture feature 7 (Figure 6, middle right) that differentiates between nuclei with smooth and rather grainy texture. Cell shape feature 14 (Figure 6, lower left) has its minimal value found in beaker-shaped cells with smooth surface and its maximal value in rather round cells with extremely rugged surface, and is specific for ciliary band cells (cluster 15.2). Foregut muscles (cluster 13) can be described using nuclear shape feature 66 (Figure 6, lower right), having one extreme in nuclei with a small compartment with a rough surface (as a result of a segmentation error) and the other extreme in elongated flat nuclei.”

6) In figure 1 b/c: The difference between B and lower part of C is unclear. If seen as a description of the two losses, the fact that the contrastive loss is shown twice is confusing. If seen as a description of the whole training setup, the omission of the fine-grained features is the issue.

We thank the reviewers for spotting this ambiguity. We omitted fine texture features since the training procedure for coarse and fine texture is the same. We have now made it explicit in the figure and its caption:

“Training procedure for the coarse and fine texture features (here illustrated by coarse texture).”

7) In figure 2: It would be interesting to find out which subset of features correlates with which class, and whether those are meaningful. At minimum, knowing whether a shape, coarse texture, and fine texture are all involved in predictions.

We agree with the reviewers that such an analysis is indeed interesting. Before the paper submission we already attempted tracing the classifier's decisions to specific features doing various feature importance analyses. However, this proved to be challenging, since the features are highly correlated not just within the ‘morphology components’ (e.g. fine nuclear texture or cell shape), but also between these components. This makes sense, because most of the morphology features do correlate for biological reasons: extremely long cells (muscles) will always have contractile fibres in their fine texture, cell shape will correlate with nuclear shape, etc. However, this makes it barely possible to isolate single features, responsible for the classifier’s decisions. Nevertheless, to get at least a rough idea of how each morphological component contributes to the prediction accuracy, we trained the classifier on these components separately and reported the results in the new Figure 2—figure supplement 2 and in the Results section (line 191):

“To explore to which extent each of the morphology components (e.g. cell shape or fine nuclear texture) contributes to predicting the manually labelled cell classes, we ran a similar classification pipeline using these components separately (Figure 2—figure supplement 2). The results show that, for example, cytoplasm texture is sufficient to tell some classes apart with high precision. However, both coarse and fine cytoplasm texture perform slightly worse on distinguishing neurosecretory cells, in which nuclei occupy almost the whole cell volume. Cell shape is satisfactory to characterise neurons and muscle cells, but shows inferior performance on epithelial and secretory cells, which display a great variety of cell shapes. Surprisingly, nuclear fine texture features correctly distinguish, among other classes, muscle cells, suggesting that this class has characteristic chromatin texture.”

8) In figure 2: The legend on the scale bars says 'mkm', which is not an abbreviation of micrometers that I am used to. Perhaps μm instead? The legend is also difficult to see (see pt. 11).

This has been fixed.

9) In figure 5: the scale bar legend is too small to see. Also, putting it above the scale bar might improve readability.

This has been fixed.

10) In Figure 7 + text: the text suggests that the clusters have been chosen manually, rather than using community detection as in the other figures. This should be justified if true.

This was done for aesthetic reasons: to visualise cell types, not clusters on the UMAP representation. However, after the reviewers pointed it out, we do agree that it is not consistent with the figure visualising MorphoFeatures clusters. That is why we changed this figure to visualise clusters as well.

11) In figure 8B + text (p.14): There isn't much said about the cells that are misclassified in the MorphoContextFeatures, i.e. where both manual segmentation and gene clusters agree, but MorphoContextFeatures does not. For example: green cell among the brown, or yellow cells just right of the central line of the organism, top. A justification similar to the explanations of misclassification in Figure 2 would help strengthen the argument.

We thank reviewers for the suggestion. We now also describe the cases of misclassification by MorphoFeatures (line 350):

“Unaware of the physical tissue boundaries, MorphoContextFeatures sometimes led to ambiguous assignment of cells neighbouring other tissues, especially cells with strongly different morphology. Such noisy assignment can be noticed, for example, in the neurons bordering muscle cells on the boundary of the cirral ganglia and in the center of the brain.”

For future work: Currently many EM reconstructions are nervous systems of somewhat higher animals (*Drosophila* and small portions of mammal brains). The shapes of these cells are very complex, and it would be interesting to see if the morphology features will continue to work on such complex cells. *Drosophila* could be a good example.

We fully agree with the reviewers that applying the pipeline to neurons is a very interesting use case. Moreover, a recent work (Dorkenwald, 2022) has shown that applying a similar contrastive learning pipeline to the texture of segmented neurons extracts features useful to distinguish between the main glia types and general neuron classes. However, the essential component of neural identity is their branching patterns, thus, to get a full morphological description of neurons one would also need to adjust the shape learning part of our pipeline. This is, however, not trivial since our augmentations focus on the geometry of the shapes but not their topology, and are thus not appropriate for branching structures. More specifically, applying our shape deformations to neurons would deform neurite as well as somata surfaces while preserving the neurite skeletons. The resulting effect on the learned representations is difficult to estimate and would require further experiments with suitable evaluation metrics.

There is a question (line 409) of how well patch characteristics will correspond when comparing different samples. This could be tested, at least in part, by applying different image normalizations to the same sample, then treating them as two separate samples.

The approach suggested by the reviewers is interesting, however, we would argue that varying intensities are a lesser problem when comparing two EM datasets. Moreover, to some extent this problem could be alleviated by current intensity correction algorithms. We would expect more issues to arise, for example, from sample preparation and specific EM modality. Accurately simulating such differences, however, is not yet possible. Thus, we would rather investigate this direction once another comparable EM volume is available.

References

Beier, T., Pape, C., Rahaman, N., Prange, T., Berg, S., Bock, D. D., … and Hamprecht, F. A. (2017). Multicut brings automated neurite segmentation closer to human performance. *Nature methods*, *14*(2), 101-102.

Dorkenwald, S., Li, P., Januszewski, M., Berger, D. R., Maitin-Shepard, J., Bodor, A. L., … and Jain, V. (2022). Multi-Layered Maps of Neuropil with Segmentation-Guided Contrastive Learning. *bioRxiv*.

Pape, C., Matskevych, A., Wolny, A., Hennies, J., Mizzon, G., Louveaux, M., … and Kreshuk, A. (2019). Leveraging domain knowledge to improve microscopy image segmentation with lifted multicuts. *Frontiers in Computer Science*, 6.

Vergara, H. M., Pape, C., Meechan, K. I., Zinchenko, V., Genoud, C., Wanner, A. A., … and Arendt, D. (2021). Whole-body integration of gene expression and single-cell morphology. *Cell*, *184*(18), 4819-4837.